# Therapeutic candidates for keloid scars identified by qualitative review of scratch assay research for wound healing

**Mohammadali E. Alishahedani[1], Manoj Yadav[1], Katelyn J. McCann[2], Portia Gough[1], Carlos R. Castillo[1], Jobel Matriz[1], Ian A. Myles**[1]*

1 Epithelial Therapeutics Unit, Laboratory of Clinical Immunology and Microbiology, National Institute of Allergy and Infectious Diseases, National Institutes of Health, North Bethesda, Maryland, United States of America, 2 Laboratory of Clinical Immunology and Microbiology, NIAID, NIH, Bethesda, MD, United States of America

* mylesi@niaid.nih.gov

**Data Availability Statement:** All relevant data are within the manuscript and its Supporting Information files.

## Abstract

The scratch assay is an *in vitro* technique used to analyze cell migration, proliferation, and cell-to-cell interaction. In the assay, cells are grown to confluence and then 'scratched' with a sterile instrument. For the cells in the leading edge, the resulting polarity induces migration and proliferation in attempt to 'heal' the modeled wound. Keloid scars are known to have an accelerated wound closure phenotype in the scratch assay, representing an overactivation of wound healing. We performed a qualitative review of the recent literature searching for inhibitors of scratch assay activity that were already available in topical formulations under the hypothesis that such compounds may offer therapeutic potential in keloid treatment. Although several shortcomings in the scratch assay literature were identified, caffeine and allicin successfully inhibited the scratch assay closure and inflammatory abnormalities in the commercially available keloid fibroblast cell line. Caffeine and allicin also impacted ATP production in keloid cells, most notably with inhibition of non-mitochondrial oxygen consumption. The traditional Chinese medicine, shikonin, was also successful in inhibiting scratch closure but displayed less dramatic impacts on metabolism. Together, our results partially summarize the strengths and limitations of current scratch assay literature and suggest clinical assessment of the therapeutic potential for these identified compounds against keloid scars may be warranted.

## Introduction

The scratch assay is an *in vitro* technique used to analyze cell migration, proliferation, and cell-to-cell interaction. In the assay, cells are grown to confluence and then 'scratched' with a sterile instrument. For the cells in the leading edge, the resulting polarity induces migration and proliferation to attempt to 'heal' the modeled wound [1]. Keloids represent a disordered scar formation marked by an overactivation of proliferation and migration, which may represent overactivation of epithelial-to-mesenchymal transition (EMT) [2–4]. Although EMT is definitionally limited to epithelial derived cells such as keratinocytes (KC), scratch assay closure

**Funding:** This work was supported by the Intramural Research Program of the National Institute of Allergy and Infectious Diseases (NIAID) and the National Institutes of Health (NIH).

**Competing interests:** The authors have declared that no competing interests exist.

time can be assessed in the scratch assay across varying cell types [5]. Given that the established treatments for keloid scars corticosteroids and 5-fluorouracil both inhibit cellular proliferation and migration [6,7], we hypothesized that systematically reviewing the current literature might identify known inhibitors of scratch assay healing times which present therapeutic potential for patients with keloid scars. We thus performed a qualitative review of the recent literature searching for inhibitors of scratch assay activity that were already available in topical formulations.

Although several shortcomings in the scratch assay literature were identified, caffeine and allicin successfully inhibited the abnormalities of proliferation/migration in the commercially available keloid fibroblast cell line compared to the healthy volunteer cell line control. Allicin also inhibited production of the inflammatory mediator interleukin (IL-) 6. Caffeine and allicin treatment inhibited mitochondrial oxidative phosphorylation (OxPhos), which worsened the inherent defect in in keloid cells. However, treatment with the mitochondrial ATP inhibitor rotenone failed to inhibit scratch closure and suggested that caffeine and allicin may exert influence through their effect on non-mitochondrial ATP production. The traditional Chinese medicine shikonin was also successful in inhibiting scratch closure but displayed less dramatic impacts on metabolism. Collectively, our results partially summarize the strengths and limitations of current scratch assay literature and suggest clinical assessment of the therapeutic potential for these identified compounds against keloid scars may be warranted.

## Methods

This work was approved by the IRB of the National Institutes of Health.

### Qualitative literature assessment

The quoted phrase "scratch Assay" and "scratch wound assays" were searched in PubMed on 5/1/2019. Search results were limited to those published on 1/1/2016 or later. A team of three reviewers (AA, CC, and IM) performed title and abstract level review to eliminate papers focusing on either neoplastic metastasis or those recommending changes to scratch assay methods. Remaining papers were read in detail to assess the use of the scratch assay, the cell types employed, and the impact of the stimulants or challenges tested.

### Cell cultures and scratch assay

All cells used in this study were purchased from commercial biobanks (www.atcc.org) or Thermo Fisher Addexbio. Cell lines were not collected for this study, were collected through medically prescribed processes, and were completely de-identified to the researchers before access. Human primary fibroblasts (ATCC PCS-201-012) and primary human keloid fibroblasts (ATCC CRL-1762) were purchased from American Tissue Culture Collection (ATCC; Manassas, VA). HaCaT keratinocytes were purchased from Thermo Fisher Science (Waltham, MA). All cells were cultured and proliferated as previously described [8]. 96 or 24 well plates (Corning; Corning, NY) were coated with 1mg/mL rat tail collagen (Roche; Basel, Switzerland) overnight at 4˚C. For 96 well plates, 17,000–25,000 cells and for 24-well plates 100,000–150,000 cells were added and allowed to adhere to the culture plate (cell number was matched across conditions within each given experiment). 12–24 hours later cells were scratched using the Autoscratch (BioTek; Winooski, VT). Cells were placed in the Cytation 5 (BioTek) at 37˚C with 5% CO2; images and enumeration were performed by the Scratch App (BioTek). Chondroitin from shark cartilage, caffeine, and folic acid were purchased from Sigma (St Louis, MO). HaCaT cells were cultured in Defined Keratinocyte-Serum Free Media (Gibco,

ThermoFisher) in T75 flasks. Once cells reached 80% confluence, they were trypsinized and seeded into a 24-well plate at 150,000 cells/well prior to undergoing the scratch assay as above.

## Immunofluorescence staining

Cells were fixed with 4% paraformaldehyde (PFA; Cat. No. 15710; Electron Microscopy Sciences, Hatfield, PA) for 20 minutes. After fixation, cells were processed for the immunostaining protocol. Cells were washed with 1X PBS for three times 5 minutes each to get rid of the PFA solution. Cells were permeabilized with 0.5% Triton X-100 (T8787-100ML; Sigma-Aldrich) solution for 15 minutes. Followed by a PBS wash and cells were blocked with 5% normal goat serum (Cat. No. 50062Z; Thermo Fisher Scientific) for 60 minutes. Rabbit Anti-Vimentin antibody (Cat. No. #5741; Cell Signaling Technology, Danvers, MA) solution was prepared in the 1:1 PBS and normal goat serum solution. Primary antibody in 1:500 dilution was incubated for 60 minutes at room temperature. Followed by three PBS wash for 5 minutes each to remove the unbound antibody. Cells were incubated with anti-Rabbit-568 Alexa flour secondary antibody (Cat. No. A-11034; Thermo Fisher Scientific) solution at 1:750 dilution in phosphate buffered saline (PBS) and normal goat serum solution for 30 minutes. Followed by three PBS wash 5 minutes each to remove the unbound secondary antibody. Next cells were stained with DAPI solution (Cat. No. 62248; Thermo Fisher Scientific) 1:2000 dilution in PBS for 30 minutes at room temperature. Followed by cells were washed four time with PBS solution 5 minutes each to remove the unbound DAPI solution. After that cells were imaged with Cytation 5 fluorescence microscope (BioTek). All the images were analyzed and processed with Gen5 software (BioTek).

## Multiplex for chemokines and cytokines

Multiplex cytokines and chemokines were performed using the Bio-plex kits per manufacturer instructions (Bio-RAD; Hercules, CA).

## Seahorse

Cellular oxidative phosphorylation (OXPHOS) and glycolysis were measured using the Seahorse Bioscience Extracellular Flux Analyzer (XFe96, Seahorse Bioscience Inc., North Billerica, MA, USA) by measuring oxygen consumption reate (OCR; indicative of respiration) and extracellular acidification rate (ECAR; indicative of glycolysis) in real time according to manufacturer's protocol.

Briefly, 10,000 fibroblasts from the healthy volunteer cell line, keloid patients were seeded in 96-well cell culture microplates designed for XFe96 in 200 µl of appropriate growth media. Fibroblasts were cultured with various stimuli for 2 hours. Prior to measurements, growth media was removed and replaced with 180 µl pH ready Seahorse Assay Media (Agilent; Catalog #103575–100) and incubated in the absence of CO2 for 1 hour in the Biotek Cytation1 instrument during which time pre-assay brightfield images were collected. Cells were sequentially treated with oligomycin (2 µM), carbonyl cyanide-4-(trifluoromethoxy)phenylhydrazone (FCCP) (0.5 µM), and rotenone + Antimycin A (0.5 µM). OCR and ECAR were then measured in a standard six-minute cycle of mix (2 min), wait (2 min), and measure (2 min). Basal levels of OCR and ECAR were recorded first, followed by OCR and ECAR levels following injection of compounds that inhibit the respiratory mitochondrial electron transport chain, or ATP synthesis. All OCR and ECAR values were normalized following the Seahorse Normalization protocol. Briefly, after the assay cells were stained with 2µg/mL Hoechst 33342 (ThermoFisher Scientific) for 30 minutes while performing post-assay brightfield imaging. Cells were then

imaged and counted using the Biotek Cytation 1. Cell counts were calculated by Cell Imaging software (Agilent) and imported into Wave (Agilent) using the normalization function.

### Statistical analysis

To determine statistical significance, analysis of variance (ANOVA) with multiple-comparison corrections were applied using GraphPad Prism 8 software (San Diego, CA). Data are presented as the mean +/- SEM. A *P* value of less than 0.05 was considered significant.

## Results

### Most scratch assay articles evaluate metastasis rather than wound repair

A PubMed search revealed 1,331 articles contained the phrase "scratch assay" or "scratch wound assay" (Fig 1A). To focus on recent publications, we limited to those published after 2016 and found 859. Our goal was to assess tissue repair and not neoplastic metastasis; thus, title and abstract level assessment was used to eliminate 449 articles. An additional 28 articles were eliminated due to their focus on methods of the scratch assay rather than influencing wound healing pathways. The remaining 382 articles were evaluated to derive the cell type used, the stimuli employed, the impact on inhibition or enhancement of the scratch assay closure time, and the scientific short comings (Fig 1A).

### Cell types were often tested in isolation

12.8% of publications that used endothelial cells and thus were reflective of endothelial-to-mesenchymal transition (EndoMT) rather than EMT (Fig 1B). Stem cells were used in 9.1% of papers which comment on the important role EMT plays in embryologic development [9]. Of the 207 publications that used epithelial cells, keratinocytes (KC), and fibroblasts (FB) were the most common cell lines used in scratch assay analysis (Table 1; Fig 1C). Among KC, 42 of the 58 (72.4%) studies used HaCaT cells, an immortalized, aneuploid cell line from adult human skin [10]. Primary skin cells were only used in 26.3% of the studies employing KC and none of the publication we evaluated directly compared HaCaT cells to primary cultures.

It is important to note that EMT does not definitionally occur in FB cells, which are already in a mesenchymal phenotype, however FB cells can undergo the inverse process of mesenchymal to epithelial transition (MET). Therefore, scratch assay results that use FB comment on wound closure via cellular migration, proliferation, or both. FB studies however used primary cells in 57 of the 105 studies (54.3%). Tissue sourcing of the primary FB cells studied was varied; 54.4% used FB from skin, 15.8% from gingiva, 10.5% from pulmonary organs, 7% from cardiac, 3.5% from tendons, and 8.8% from other tissues (Fig 1C). 83.3% of cell lines used were from animals (Fig 1C).

### Experiments often lacked proper controls, especially for natural products

Natural products were frequently used in scratch assay experiments evaluating potential agents that could enhance wound healing in FB, KC, and epithelia (Table 1). However, control groups were often limited to diluent rather than a similar but comparable challenge. As one singular but demonstrative example, researchers showed enhanced scratch healing in HaCaT cells exposed to crocodile serum [25]. However, while the crocodile serum did demonstrate a dose response curve, no competing serum was used in the studies (such bovine serum). Therefore, it is unclear whether the findings are unique to the specific serum they used or if any serum would have similar effects as suggested by research demonstrating similar impacts of the serum-product lactoferrin [11]. There are several other examples of plant or animal extracts

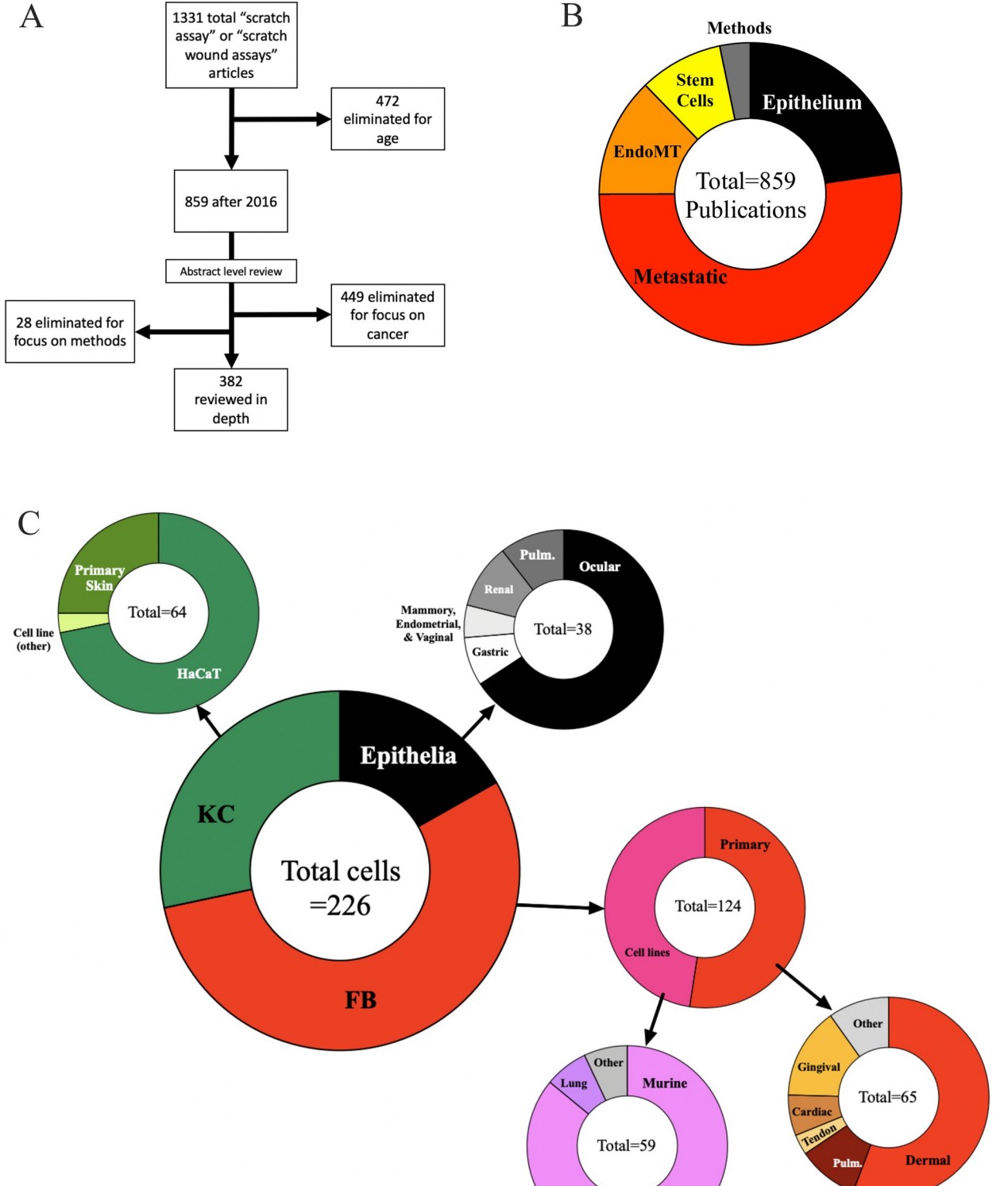

**Fig 1. Articles using the scratch assay vary by cell type.** (A) Progression and enumeration of articles found during qualitative literature review. (B) Pie chart of the total articles uncovered separated by general wound healing (often EMT) type or EndoMT. (C) Pie chart breakdown of total cell types used in the identified literature. KC = keratinocyte; FB = fibroblasts. EMT does not definitionally occur in FB, which are already of the mesenchymal phenotype.

**Table 1. Effect of different stimuli on scratch assay in keratinocytes, fibroblasts, and epithelial cells.** Migration rate of scratch assay in response to cell signalers, natural products, metabolic mediators, immune mediators, drugs, and other stimuli.

| HaCaT Keratinocytes | | |
|---|---|---|
| **Increase** | **Decrease** | **No Impact** |
| *Cell Signalers* | | |
| •EGF [11,12]<br>•Pep19-2.5 [13]<br>•Hepatocyte Growth Factor (HGF) [14]<br>•micro RNA 21 [15]<br>•ERBB2 [16]<br>•Keratinocyte Growth Factor (KGF) [17]<br>•AES16-2M (ERK activating peptide) [18]<br>•GFP-Smad2 [19]<br>•Lipofectamine and KGF-mRNA [17]<br>•SIS3 (Smad3 phosphorylation specific inhibitor) [19]<br>•Liraglutide, a Glucagon-like peptide-1 analogue (concentration dependent) [20]<br>•Thrombin [21]<br>•AHR siRNA [22] | •EGF receptor inhibitor (EGFRi) [19]<br>•JNK inhibitor (JNKi) [19]<br>•MEK1 inhibitor (MEKi) [19]<br>•Enhanced green fluorescent protein (eGFP) with Lipofectamine [17]<br>•LY 294002 (PI3K inhibitor) [20]<br>•IDR-1018, a synthetic innate defense regulator peptide, in normoxia [23]<br>•EGFR antagonist (AG1478) [21]<br>•ERK1/2 antagonist (UO126) [21]<br>•AHR antagonist, CH223191 [22]<br>•RIPK4 via TGFβ [24] | •ERBB3 [16]<br>•ERK inhibitor (U0126) [18]<br>•IDR-1018 in hypoxic condition [23]<br>•ALK5 (TGF-ß receptor I) inhibitor (TGFRi) [19]<br>•Cytochalasin D [22]<br>•PXR siRNA [22]<br>•PXR antagonist, SPA70 [22]<br>•JNK inhibitor (SP600125 and indirubin) [22] |
| *Natural Products* | | |
| •*Crocodylus siamensis* serum [25]<br>•*Vitex negundo, Emblica officinalis gaertn,* and *Tridax procumbens* mix [26]<br>•Tracheloside [27]<br>•Fish scale derived gelatin nanofibrous scaffolds [28]<br>•Quercetin (plant flavonol) [29]<br>•Aloe vera extract [30,31]<br>•Chitosan, polyvinyl alcohol S-nitroso-N-acetyl-DL-penicillamine gel (taken from eggs) [32]<br>•*Aloe purpurea Mascarene* (APM) [31]<br>•*Annona crassiflora* flavonoid seed extracts [33]<br>•glucan extract of *Ziziphus jujuba* [34]<br>•Indirubin [22] | •Chondroitin sulfate (ECM polysaccharide) [35]<br>•Caffeine [36]<br>•*Rhodomyrtone* [37]<br>•*Annona crassiflora* flavonoid peel extracts [33] | •*A. tormentorii* [31]<br>•*A. lomatophylloides* [31]<br>•*A. macra* [31]<br>•*A. purpurea* (Réunion) [31]<br>•Quercetin (dose dependent) [29]<br>•*Calendula officinalis* n-hexanic, ethanolic or aqueous extracts [14]<br>•single triterpenes (α-amyrine, β-amyrine, lupeol, taraxastene) [14]<br>•β-carotene [14]<br>•Triterpene esters [14]<br>•Tannic Acid (TA) [38] |
| *Metabolic Mediators* | | |
| •Insulin [39]<br>•Melatonin [40]<br>•Neurotensin [39]<br>•Substance P [39]<br>•Human and bovine lactoferricin [11] | | •Allantoin [27]<br>•Lactoferricin in presence of mitomycin C [11] |
| *Immune Mediators* | | |
| •IL-8 [41]<br>•poly I:C via IL-8 [41]<br>•Neutrophil extracellular traps [42]<br>•TGFβ [18] | •Chloroquine via poly I:C [41]<br>•Anti TGFβ [19] | •anti-IL-8 antibody [41] |
| *Drugs* | | |
| •Phenytoin [43]<br>•Remifentanil pretreatment (RPC) via H2O2 [44] | •Nanoemulsion [43]<br>•Mitomycin C [20,22]<br>•Rapamycin [45]<br>•H2O2 [44]<br>•3 Methyladenine (3-MA) countering Remifentanil + H2O2 [44] | •Rifampicin [22] |
| *Other* | | |
| •Amniotic membrane [19]<br>•Tannic acid (TA)-modified Silver nanoparticles (AgNPs) [38]<br>•Ag(salH)2 [46]<br>•AgNO3 [46]<br>•Stem cell media [47]<br>•human stromal vascular fraction gel [48]<br>•gallium/cerium-doped phosphate glass fibers [49]<br>•Mycosporine-like Amino acids: shinorine, porphyra-334, mycosporine-glycine-alanine, or bostrychine [50]<br>•Media with 20% fetal calf serum (FCS) vs 10% FCS [50] | | •Silver nanoparticle [51]<br>•AgNO3 (silver nitrate) [46]<br>•ozonated PBS [52] |
| **Keratinocytes Other** | | |
| **Increase** | **Decrease** | **No Impact** |
| *Cell Signalers* | | |

(*Continued*)

**Table 1.** (Continued)

| | | |
|---|---|---|
| •JWH015 (cannabinoid receptor type 2 agonist) [53] <br> •PHD-2—Protein Hydroxylase Domain Containing Protein 2 [54] <br> •IOX2 (PHD-2 inhibitor) in hypoxia [54] <br> •Epidermal growth factor (EGF) [55] <br> •CD163 overexpressing macrophages [56] <br> •Indirubin [22] | • AM281 (cannabinoid receptor type 1 antagonist) [53] <br> • AM360 (cannabinoid receptor type 2 antagonist) [53] <br> •Cldn-1 knockdown [57] <br> •ZO-1 knockdown [57] | •Ocln knockdown [57] |
| *Natural Products* | | |
| •*Spirulina extract* [58] <br> •Cholinergic acid (C) [59] <br> •Quercetin [55] <br> •Hydroxysafflor Yellow A, a derivative of safflower [60] <br> •*H. perforatum* oil extract [61] | | •Hidradenitis suppurativa (HS)[62] <br> •Myricetin-3-O-rhamnoside (M) [59] <br> •*P. percica* extract [59] |
| *Drugs* | | |
| | •Ingenol mebutate [63] <br> •Sulfur Mustard in hypoxic condition [54] | |
| *Other* | | |
| •Adipose derived stem cells in hypoxic conditions [64] <br> •Conditioned medium dermal stromal cells (cmDSCs) [65] <br> •Light Emitting Diode (LED) (significance not indicated) [66] <br> •DermaLife media [57] | •Chronic wound [62] <br> •Orofacial clefts (OFC) [67] <br> •Rodent keratinocytes versus mouse keratinocytes [68] | •conditioned medium adipose stromal cells (cmASCs) [65] <br> •EpiLife [57] |
| **Human Dermal Fibroblasts** | | |
| **Increase** | **Decrease** | **No Impact** |
| *Cell Signalers* | | |
| •EGF [69,70] <br> •Fli1 siRNA [71] <br> •siKI RNA [72] <br> •bFGF (basic fibroblast growth factor) [70] <br> •Apelin and ML-233 (apelin receptor activator) [73] <br> •Klotho (KI), gene that encodes αKI protein [72] <br> •Mono-epoxy-tocotrienol-α (MeT3α) [74] <br> •JHW015 (a cannabinoid receptor type 2 agonist) [53] <br> •PHD-2—Protein Hydroxylase Domain Containing Protein 2 [54] <br> •IOX2 (PHD-2 inhibitor) in hypoxia [54] <br> •Cartilage acidic protein 1 [75] | •AQP1 siRNA [71] <br> •AM281 (cannabinoid receptor type 1 antagonist) [53] <br> •AM360 (cannabinoid receptor type 2 antagonist) [53] <br> •Compound 21 [76] <br> •Cryptotanshinone [77] | •scrRNA [72] |
| *Natural Products* | | |
| •Exopolysaccharides (EPS) from Nitratireductor spp PRIM-31 [78] <br> •Spirulina platensis (algae) [58] <br> •*Moringa oleifera* [79] <br> •Myricetin-3-O-β-rhamnoside (M) [59] <br> •Cholinergic acid (C) [59] <br> •Saffron [80] <br> •*Triticum vulgare* extract [81] <br> •Cupuassu butter [82] <br> •*Hypermongone C* extracted [83] <br> •*Moringa oleifera* fraction [79] <br> •*C. Papaya* extract [84] | •Myricetin-3-O-β-rhamnoside [59] <br> •*P. percica* extract [59] <br> •Allicin [85] | •*Moringa oleifera* ethyl acetate fraction [79] |
| *Metabolic Mediators* | | |
| •Angiotensin 2 [76] <br> •Vitamin combinations: B9 and B12; B3, B5, B6, and B10; and B3, B5, and B7 [86] | | •Vitamin Combination: B3, B5, B6, B9, B10, and B12 [86] <br> •Vitamin C [86] |
| *Drugs* | | |
| •Estradiol [87] <br> •Pimecrolimus [88] | •Arsenic [87,89] <br> •Sulfur mustard in hypoxia [54] <br> •Mitomycin C [77] | |
| *Immune Mediators* | | |
| •IL-6 [90] <br> •CXCL-8 [90] <br> •Histatins variants (Hst1, Hst2, cyclic Hst1) [90] <br> •TGFβ [71] | | |
| *Other* | | |

(*Continued*)

**Table 1.** (Continued)

| | | |
|---|---|---|
| •Scleroderma disease [71]<br>•Conditioned medium dermal stromal cells (cmDSCs) [65]<br>•Conditioned medium adipose stromal cells (cmASCs) [65]<br>•Fibrocytes [91]<br>•Platelet-rich plasma [92]<br>•Human Adipose Derived Stem Cell (HADSC) Extracellular Vesicles (EV) [93]<br>•5% and 10% cerium chloride (CeCl3) [94]<br>•Human stromal vascular fraction gel [48]<br>•Silver nanoparticles [95]<br>•Synthetic Quanizoline Compound [96]<br>•Dermal skin cells versus adipose skin cells [65]<br>•curcumin-silica nano-particle [97] | •Chlorogenic acid [59]<br>•Depleted uranium [92] | •Serum free medium and BSA [93] |

**Human Primary Fibroblasts, Other**

| Increase | Decrease | No Impact |
|---|---|---|
| *Cell Signalers* | | |
| •TRAM 34 (K+ channel 3.1 inhibitor) [98]<br>•Fibrocytes [91]<br>•JWH015 (cannabinoid rec agonist) [53]<br>•Irisin [99]<br>•FKBP10 KD via TGFβ1 [100]<br>•PDRN (polydeoxyribonucleotide) [101]<br>•Enamel matrix proteins (EMPs) [102]<br>•miRNA-34a inhibitor [103]<br>•MiRNA-34a and delta-like protein 1 (DLL1) siRNA [103] | •Human amniotic epithelial cells [104]<br>•TRAM 34 (calcium/calmodulin activated K+ channel 3.1 inhibitor) via TGFb [98]<br>•FK506-binding protein 10 (FKBP10) knockdown [100]<br>•ERK inhibitor via IL-25 [105]<br>•SB (p38 inhibitor) via IL-25 [105]<br>•SP (JNK inhibitor) via IL-25 [105]<br>•Bay (NFκB inhibitor) via IL-25 [105]<br>•Ullrich congenital muscular dystrophy [106]<br>•Anti-collagen VI 3C4 antibody (3C4-PA) [106]<br>•SB203580 (p38 MAPK inhibitor) [107]<br>•NS398 (COX-2-specific inhibitor) [107]<br>•Heat-shock protein 27 (Hsp27) siRNA [107]<br>•MicroRNA 34 (miR-34a) mimic [103] | •Enhanced green fluorescent protein (eGFP) [17] |
| *Natural Products* | | |
| •Polydeoxyribonucleotides-a fragmented DNA from (Oncorhynchus mykiss) sperm [69]<br>•Maltodextrin/ascorbic acid [108]<br>•Coumestrol/hydroxypropyl-β-cyclodextrin [109]<br>•Hypromellose (HMPC) [109]<br>•Platelet-rich plasma (PRP) [110]<br>•Platelet-rich plasma (PRP) and fibrin [111]<br>•Indirubin [112]<br>•Ozone [113] | •*Eonurine* extract [114]<br>•*H. italicum* [115] | |
| *Immune Mediators* | | |
| •IL-25 [105]<br>•TNF-α [107] | •IL-37 [116] | •IL-1β [101]<br>•TNF-α in presence of HIF [117] |
| *Metabolic Mediators* | | |
| •Bradykinin [118]<br>•Insulin w/o adipocytes [119] | •2DG in cells from patients with rheumatoid arthritis [120]<br>•3-bromopyruvate in cells from patients with rheumatoid arthritis [120] | |
| *Drugs* | | |
| | •Pirfenidone [121]<br>•anti-IL 6 [112]<br>•anti-IL 8 [112] | |
| *Other* | | |
| •Adipocytes from non-diabetics (ND) [119]<br>•Biodentine [122] | •All trans retinoic acid [123]<br>•Zoledronic acid [111]<br>•Ullrich congenital muscular dystrophy (UCMD) [106]<br>•anti-collagen VI 3C4 antibody (3C4-PA) [106] •Oxygenating therapeutic (Ox66TM) [124] | •1% FBS vs 10% FBS in DMEM [108]<br>•Cobalt chloride (CoCl2) [102]<br>•TheraCal [122]<br>•Xeno III [122] |

**Animal Fibroblasts**

| Increase | Decrease | No Impact |
|---|---|---|
| *Cell Signalers* | | |

(*Continued*)

**Table 1.** (Continued)

| | | |
|---|---|---|
| •Apelin and ML-233 (apelin receptor activator) [73]<br>•High affinity small peptide ligand, H1 [125]<br>•Endodontic paste [126]<br>•PDGF [127–130]<br>•Protease-activated receptor-4 activating peptide (PAR-4AP) [131]<br>•Thrombin [131]<br>•Bioinspired hydrogels with basic fibroblast growth factor [132]<br>•ATP [133]<br>•microRNA 103 mimic [134]<br>•synthetic peptide SVVYGLR [135]<br>•PDGF [136]<br>•cGAMP [137] | •Apln (apelin peptide—a G protein couple receptor) siRNA [73]<br>•apelin receptor (aplnr) siRNA [73]<br>•knockdown of tRNA selenocysteine 1 associated protein 1 (Trnau1ap) [138]<br>•NS398 (COX-2 inhibitor) [139]<br>•DKK-1 (Wnt/β-catenin antagonist) [139]<br>•JNK inhibitor (SP600125) [140]<br>•SP600125 and Bay [140]<br>•PI3K inhibitor (LY294002) [140]<br>•DALBK (Bradykinin rec 1 antagonists) [141]<br>•HOE (Bradykinin rec 2 antagonists) [141]<br>•PDTC (NFkB receptor inhibitor) [141]<br>•PK (Bradykinin rec 2 antagonists) [130]<br>•miRNA-103 inhibitor [134]<br>•SPHK1 [134]<br>•Platelet derived growth factor [142] | •SF in presence of Bay 11–7082, NF-B inhibitor [143] |
| *Natural Products* | | |
| •*Thunnus obesus* (big eye tuna) extract [144]<br>•3-epimasticadienolic acid (pistachio hull extract) [145]<br>•*Pistacia vera L.* hull extract (select fractions) [145]<br>•n-butanol [145]<br>•*Talaromyces purpureogenus* (fungus) silver nanoparticles [146] (significance not indicated)<br>•Sea bass extract [147]<br>•*Achemilla vulgaris* extract [148]<br>•polyherbal formulation of *Vitex negundo, Emblica officinalis Gaertn, Tridax procumbens* [26]<br>•Cipladine (iodine cream) [149]<br>•*A. Sacata* leaf extract (significance not indicated) [149]<br>•terpinolene, α-phellandrene (monoterpenes) [127]<br>•grape seed extract [150]<br>•*Eugenia dysenterica* (*Myrtaceae*) oil (significance not indicated) [151]<br>•Prangos ferulacea roots extract [152]<br>•*Terminalia sericea* extracts [128]<br>•S. nux-vomica-ZnO nanocomposite [153]<br>•*Lafoensia pacari* leaf [129]<br>•Vegetable oil blend [130]<br>•Flavonoid extract oil palm leaf [154]<br>•*Allophylus spicatus* [155]<br>•*Ocimum gratissimum* [155]<br>•*Jasminum dichotomum* [155]<br>•*Phioliota nameko* [156]<br>•*H. perforatum* oil extract [61]<br>•Punica granatum and polymeric film [157]<br>•Triticum aestivum extract with chitosan [158]<br>•Sorocea guilleminina Gaudich extract [159] | •Ethyl Acetate Fractions of *Allophylus spicatus* [142]<br>•Ethyl Acetate Fractions of *Ocimum gratissimum* [142]<br>•Ethyl Acetate Fractions of *Jasminum dichotomum* [142] | •*Struthanthus vulgaris* extract [136]<br>•*Pistacia vera L.* hull extract (select fractions) [145]<br>•Ozonated PBS [52]<br>•Ozone therapy for wound healing [52]<br>•*Philenoptera cyanescens* folium cum fructus extract [155]<br>•*Melanthera scandens* herba extract [155]<br>•*Annona senegalensis* extract from folium [155]<br>•*Cissus quadrangularis L* extract from herba [155]<br>•*Gymnanthemum coloratum* extract from radix and folium [155]<br>•*Indigofera pulchra* extract from herba [155]<br>•*Leonotis nepetifolia* var. africana extract from herba [155]<br>•*Millettia thonningii* extract from cortex [155]<br>•*Rourea coccinea* extract from radix and folium [155]<br>•*Thonningia sanguinea* extract from herba [155]<br>•*Trichilia monadelpha* extract from cortex [155]<br>•*Triumfetta rhomboidea* extract from radix [155]<br>•*Uvaria ovata* extract from radix, cortex, and folium [155]<br>•*Ocimum gratissimum* extract from herba [155]<br>•*Jasminum dichotomum* extract [155] |
| *Metabolic Mediators* | | |
| •Allantoin [145]<br>•Bradykinin [141] | | •Insulin [119]<br>•Subcutaneous adipocytes [119] |
| *Immune Mediators* | | |
| •IL-6 [160]<br>•Lipopolysaccharide [139]<br>•TGFβ1 [161]<br>•TNF [141]<br>• Bay 11–7082 (NFkB inhibitor) [140] | •NALP3KO via ATP [133]<br>•High lung macrophage MHCII expression [162]<br>•low MHCII + PDGF-AA blocking antibody [162]<br>•Lipopolysaccharide [140]<br>•TNF antibody [141]<br>•Bay 11–7082 (NFkB inhibitor) [143] | |
| *Drugs* | | |
| •Chloroform [145] | •NaOCl sodium hypochlorite [163] | |
| *Other* | | |

(*Continued*)

**Table 1.** (Continued)

| | | |
|---|---|---|
| •Silk fibroin [143]<br>•Chitosan, polyvinyl alcohol S-nitroso-N-acetyl-DL-penicillamine gel (taken from eggs) [32]<br>•Silver nanoparticles [51,164]<br>•Cipladine (iodine cream) [149]<br>•Induced pluripotent stem cell-derived exosomes [165]<br>•Self-assembled Graphene Quantum Dots (sGQDs) [150]<br>•Electrical stimulation [166]<br>•Poly(2-hydroxyethyl methacrylate)/polypyrrole hydrogel [166]<br>•Sponges w/carboxymethyl chitosan and collagen peptides [167]<br>•Gold nanoparticles (AuNP) [168]<br>•Stromal vascular fraction (SVF) [169]<br>•Quinone-based chromenopyrazole (QCP) antioxidant-laden silk fibroin electrospun nanofiber scaffold [170]<br>•Pulmonary fibrosis associated RNA overexpression [171]<br>•Titanium dioxide nanoparticle biofilm [172]<br>•Neonatal cardiac fibroblasts infected with ETV2 [173]<br>•exosomes platelet rich plasma [174]<br>•tonsil derived Stem cell media [175]<br>•gallium/cerium-doped phosphate glass fibers [49] | •iodoform-based paste [126]<br>•Tp-AgNPs [146]<br>•SB203580 (p38 MAPK inhibitor) [160]<br>•Ferrous nanoparticles [176]<br>•Elaidic and linoleic (fatty acids) [119]<br>•Light exposure [177]<br>•Primary rat alveolar macrophages (AMO)-derived monocyte chemotactic protein-induced protein 1 (MCPIP1) knockdown [178]<br>•Oxymatrine and Notch signaling pathway inhibitor (DAPT) via TGF-β1 [161] | •Ca(OCl2) (calcium hypochlorite) [163]<br>•Stromal vascular fraction in normoglycemia [169]<br>•VEGF factor E (VEGF-E) [179] |

**Epithelial Cells**

| Increase | Decrease | No Impact |
|---|---|---|
| *Cell Signalers* | | |
| •Epidermal Growth Factor [12]<br>•tBHQ (Nrf2 inducer) [180]<br>•Erythroid E2-related factor 2 (Nrf2) [180]<br>•siRNA-knockdown of epiplakin [181]<br>•IWR-1 (Wnt inhibitor) [182]<br>•Platelet-derived growth factor isoform BB (PDGFBB) [183]<br>•Corneal Mesenchymal stromal cells exomes [184]<br>•VEGF [185]<br>•ATF 2 and ATF 7 [186]<br>•Chitinase-like protein YKL-40 [187] | •Tankyrase inhibitor XAV939 via TGF-β [188]<br>•Placental growth factor via hypoxia [189]<br>•Lamin A/C (LMNA) knockdown [190]<br>•neonatal Fc receptor [191] | •Myocardin-related transcription factor A (MRTF-A) signaling inhibitor CCG-1423 [192]<br>•miR-363 [182]<br>•PGF [189] |
| *Natural Products* | | |
| •*Centell asiatica* extracts [193]<br>•Crocetin (antioxidant carotenoid in saffron) via PDGFBB [183]<br>•Pentacyclic triterpene–rich *Centella* extract [193]<br>•Asiaticoside *Centella* extract [193]<br>•Madecassoside *Centella* extract [193]<br>•*Lactobacillus crispatus* [185]<br>•*L. crispatus* supernatent [185] | •*Fucus distichus* subspecies evanescens extract [194]<br>•Casein hydrolysates (concentration and fraction dependent) [195] | •non silencing siRNA [196]<br>•Heat killed *L. crispatus* [185]<br>•*L.acidophilus* [185] |
| *Metabolic Mediators* | | |
| •Aqueos lysophosphatidic acid [197]<br>•Hypokalemia [198] | •Estradiol [199] | •Pepsin [200]<br>•1,25-dihydroxy vitamin D3 [201]<br>•Substance P [202]<br>•All-trans retinoic acid receptor agonist [203]<br>•BMS493 (retinoic acid antagonist) [203] |
| *Immune Mediators* | | |
| •NLRP siRNA [204]<br>•IL1-β [205]<br>•NLRP3 siRNA [204]<br>•TGFβ [205] | •*S. aureus* [206] | •mTOR-siRNA [196]<br>•Hla mutant *S. aureus* [206] |
| *Drugs* | | |
| | •Bevacizumab [207]<br>•Canakinumab via TGFβ and IL1β [205]<br>•Aflibercept/Ranibizumab [208] | •Axitinib [209] |
| *Other* | | |
| •Stromal Fibroblasts Conditioned Mediums (SFCM) [202]<br>•Low sheer stress induced to HCEC before scratching [210]<br>•Differentiation in bronchial cells [211] | •Plumbagin [212]<br>•Biomass fuel smoke extract [204]<br>•Cigarette smoke extract [204]<br>•Cigarette smoke condensate [213]<br>•Crocetin via PDGFBB [183] | •Intermittent Hypoxia [214]<br>•Hypoxia [189]<br>•Amphiphilic block polymer polyethylene glycol-polycaprolactone [209]<br>•Low-intensity pulsed ultrasound [215]<br>•Particulate Material (PM2.5) (significance not indicated) [216]<br>•Sunitinib malate loaded with biocompatible poly (lactic-co-glycolic acid) nanoparticles [217] |

being tested against diluent alone rather than a competing challenge of similar, but distinct, molecular complexity. Overall, such a limitation may not have any practical implications given that a product that induces wound healing may be beneficial regardless of the mechanism. However, it does present a limitation on mechanistic insights since, for example, the effects of crocodile serum may be due the added nutrient density of the culture media. Mechanistic interpretations are easier for studies where the only variable is the addition of one cytokine or molecule than the studies where a highly complex stimulant is compared to water or saline.

### Findings were limited in mechanistic validity

One of the limitations in scratch assay publications included the failure to use objective statistical methods to evaluate results. For example, some papers would rely on photographs of healed scratches but not offer measurements of impact beyond visual comparisons [112]. In addition, some publications failed to experimentally block the pathway claimed as mechanistic. As one example, researchers suggested nicotine induced scratch closure through modulation of αSMA (alpha smooth muscle actin; a potential EMT modulator) but did not experimentally block or neutralize αSMA to validate the mechanistic claims [218].

### Commercially available inhibitors of the scratch assay were identified

Despite the limitations in the literature, we aimed to identify potential candidates for topical products that could inhibit scratch closure. Our criteria were to identify treatments that were: (a) already through the drug development pipeline; (b) available in a topical formulation; and (c) present a reasonable side effect profile. Based on these criteria we identified chondroitin [35], caffeine [36], and allicin [85] as potential scratch assay closure inhibitors.

### Chondroitin failed to inhibit scratch assay results

In direct contrast with the prior report [35] chondroitin enhanced scratch closure in heathy volunteer (HV) primary fibroblast line cells (FB; Fig 2A) and HaCaT keratinocytes (Fig 2B and 2C). Of note, the prior report using chondroitin extracted it directly from pig trachea [35] whereas our assay used pharmaceutical-grade chondroitin from shark cartilage. Given the failure to inhibit scratch closure, chondroitin was not evaluated further.

### Caffeine inhibited scratch closure in healthy and keloid cell line fibroblasts

In contrast to chondroitin, caffeine recapitulated the literature through inhibition of the scratch assay in a dose dependent manner (Fig 2D). While inhibition of wound healing pathways would be most often viewed negatively, such inhibition may be beneficial in patients with keloid fibroblasts which display pathologic overactivity of wound healing [2–4]. Similar to the prior research [5,219], we found that keloid fibroblasts demonstrated increased closure over time in the scratch assay (Fig 2E and 2F; area under the curve HV-FB = 654, 95%CI 612.5–690.3; area under the curve KEL-FB = 829, 95%CI 778.5–879.5). At lower concentrations, caffeine inhibited scratch repair in HV-FB more than for keloid FB (Fig 2G), however at concentrations above 1mg/mL equivalent inhibition was seen (Fig 2H).

Consistent with prior reports in KC [220], keloid FB had significantly more immunofluorescent staining per cell for the mesenchymal phenotype marker vimentin. Vimentin cellular expression was inhibited by caffeine (Fig 3A and 3B). To elucidate how caffeine may be influencing mesenchymal phenotypes in these cells we first evaluated the impact on cytokines previously associated with keloid scars [221]. Supernatant from keloid FB accumulated significantly more TGFβ2, but not TGFβ1 or TGFβ3 (Fig 3C–3E). However, caffeine did not

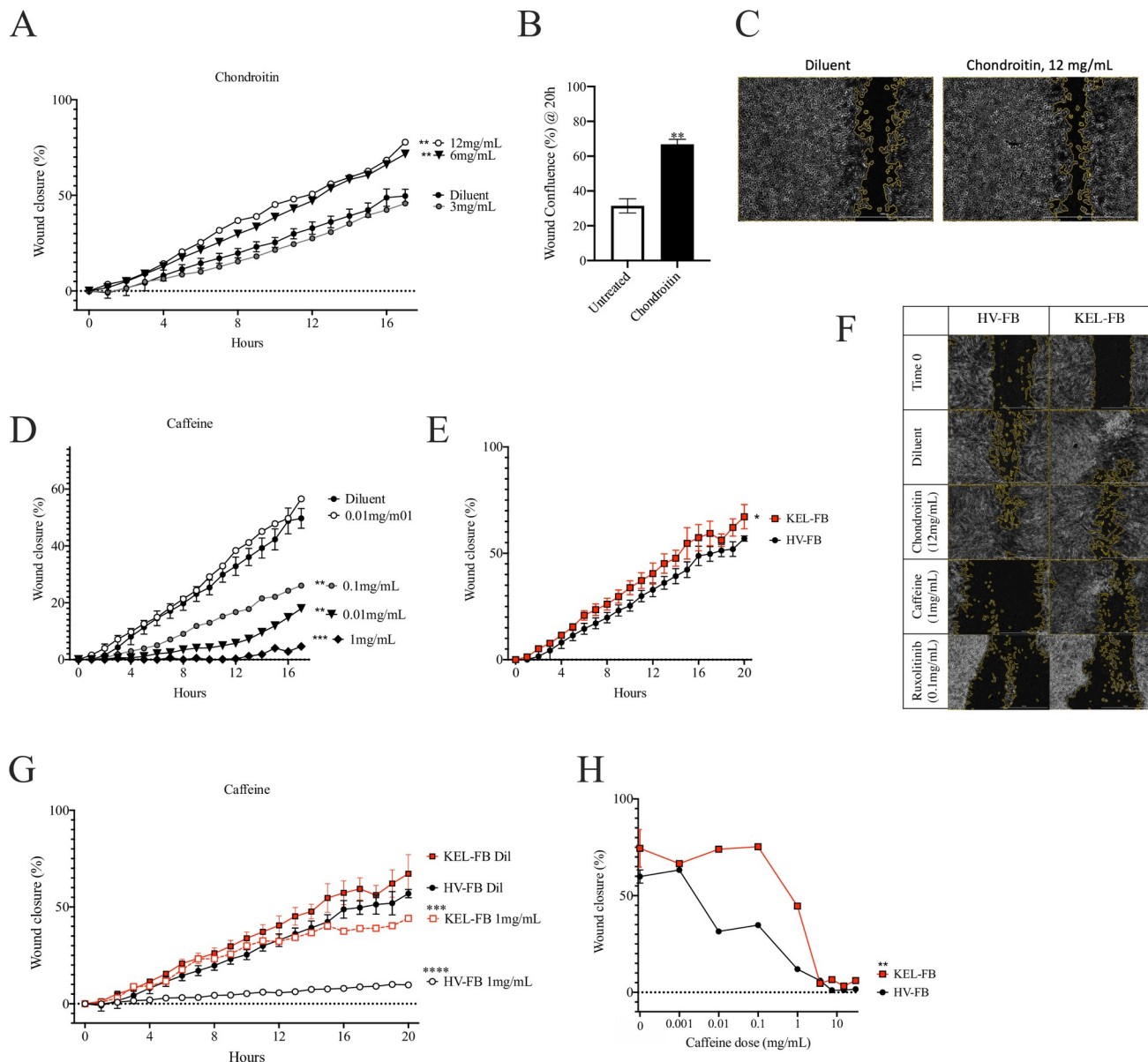

**Fig 2. Caffeine, but not chondroitin, inhibit scratch repair in keloid and healthy volunteer fibroblasts.** (A) Wound closure over 18 hours for healthy volunteer fibroblast line (HV-FB) treated with indicated concentrations of chondroitin. (B-C) Wound closure (B) and representative image (C) for HaCaT keratinocytes at 20hours with treatment of 12mg/mL of chondroitin at 18 hours. (D) Wound closure over 18 hours for HV-FB treated with indicated concentrations of caffeine. (E-F) Wound closure over 22 hours and representative images at 22 hours (F) for HV-FB or a commercially available fibroblasts cell line from a patient with keloid scaring (KEL-FB). (G) Wound closure over 22 hours for HV-FB and KEL-FB treated with 1mg/mL caffeine. (H) Wound closure at 14 hours for HV-FB and KEL-FB treated with indicated concentrations of caffeine. Results are representative of three independent experiments and displayed as mean + SEM for triplicate wells. * = p<0.05; ** = p<0.01; *** = p <0.001; for statistical comparison of area under the curve versus HV-FB under diluent stimulation conditions as determined by ANOVA with Sidak adjustment. Masking on representative images of scratch assay performed by Scratch App (BioTek).

significantly influence production of any TGFβ isotype in keloid FB (Fig 3C–3E). Keloid fibroblasts also displayed an increased supernatant accumulation of interleukin (IL-) 6 (Fig 3F) and a reduced production of IL-8 (Fig 3G). However, caffeine did not correct these abnormalities, nor did it significantly influence production of RANTES (Fig 3H), CXCL1 (Fig 3I), hepatocyte growth factor (HGF; S1A Fig), CCL2 (S1B Fig), vascular endothelial growth factor (VEGF; S1C Fig), or platelet derived growth factor (PDGF; S1D Fig).

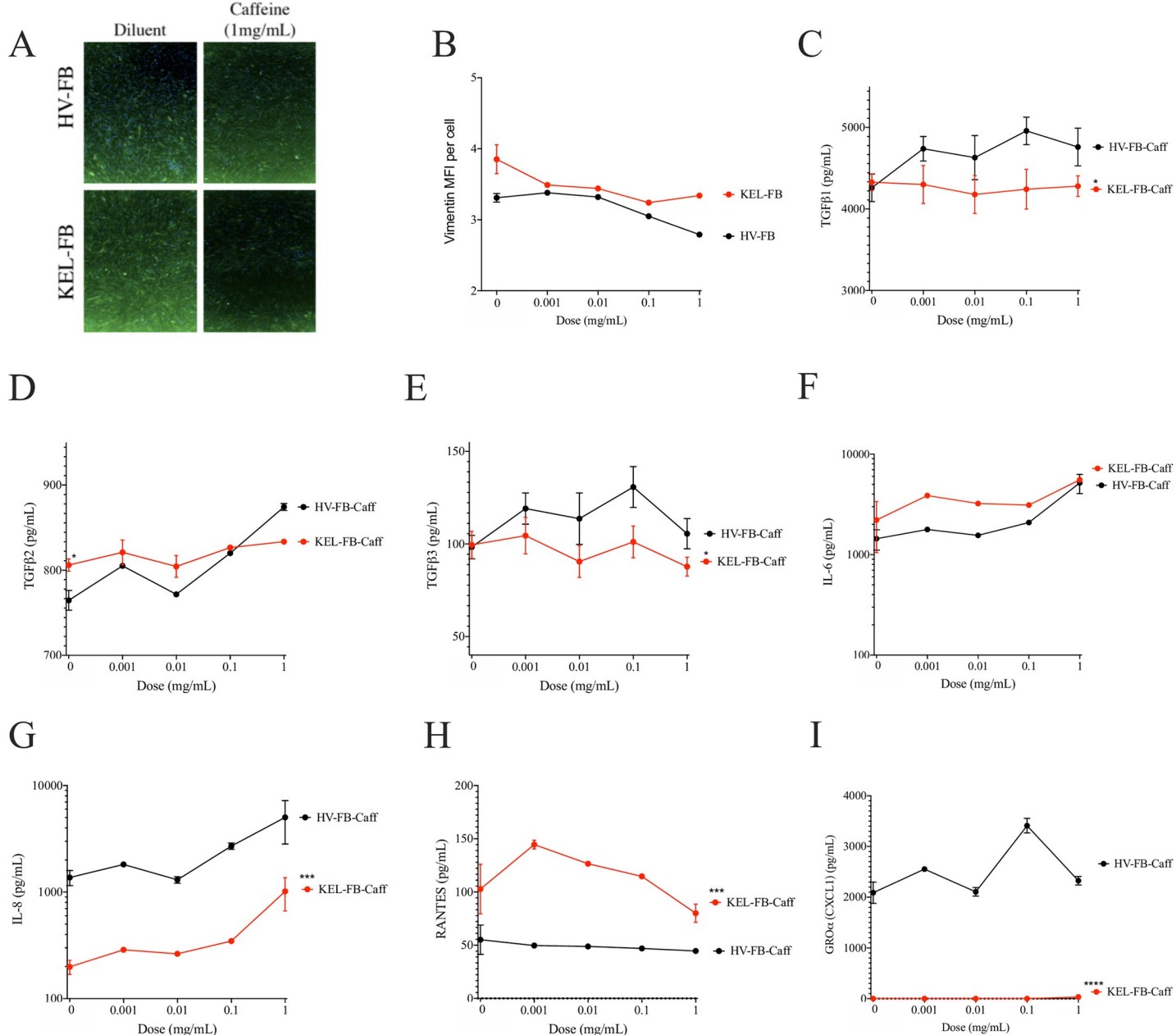

**Fig 3. Impact of caffeine on vimentin expression and cytokine production.** (A) Representative images from vimentin stain in healthy volunteer (HV-) or keloid-derived (KEL-) fibroblast cell lines (FB). (B) Mean fluorescence intensity (MFI) per cell for vimentin for HV-FB and KEL-FB treated with indicated doses of caffeine. (C-I) Supernatant accumulation of cytokines and chemokines for cells treated with caffeine. Transforming growth factor beta 1 (TGFβ1; C), TGFβ2 (D), TGFβ3 (E), interleukin (IL-) 6 (F), IL-8 (G), RANTES (H), and CXCL1/GROα (I) are shown. Results are representative of two independent experiments and displayed as mean + SEM for triplicate wells. $^{*}$ = p<0.05; $^{***}$ = p <0.001; for statistical comparison of area under the curve versus HV-FB under diluent stimulation conditions as determined by ANOVA with Sidak adjustment.

## Caffeine altered metabolic activity in keloid fibroblasts

Cells form keloid scars are known to display Warburg metabolism–a form of metabolic disruption associated with cancer cells in which over expression of STAT3, in conjunction with JAK2, drives a tendency for rapidly proliferating cells to generate ATP via glycolysis rather than OxPhos, even with available oxygen [222,223]. Consistent with these reports, keloid

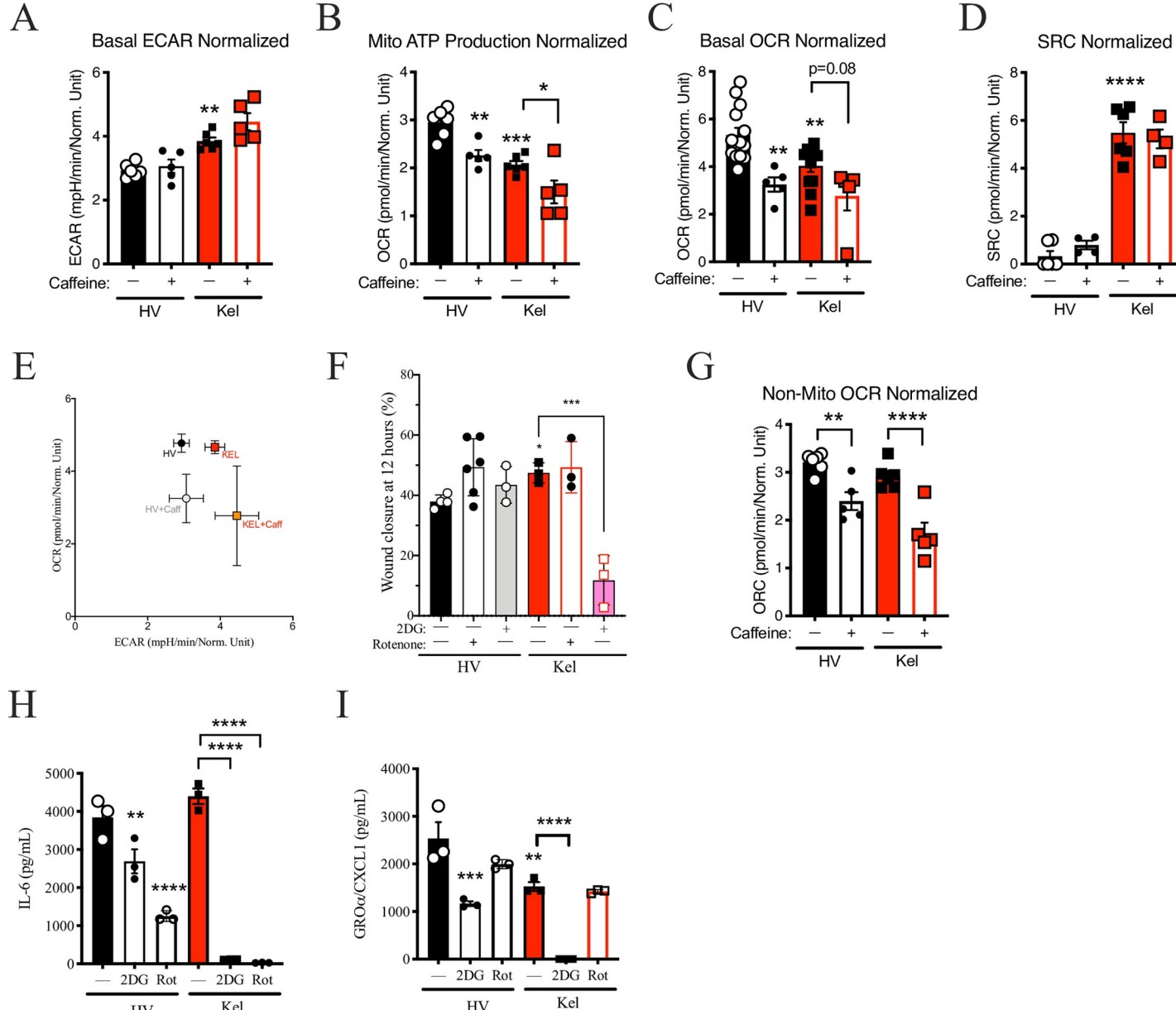

**Fig 4. Caffeine impacts metabolic function.** Seahorse assay was performed on fibroblast cell lines (FB) from healthy volunteer (HV) or keloid scars (Kel). Results for extracellular acidification rate (ECAR; a measure of glycolysis; A), mitochondrial (mito) ATP production (B), basal oxygen consumption rate (OCR; C), spare respiratory capacity (SRC; D), and ratio of basal ECAR to OCR (E) are shown. (F) Wound closure at 12 hours for keloid FB or healthy FB with treatment with diluent, the glycolysis inhibitor 2DG, or the mitochondrial OxPhos inhibitor rotenone. (G) Seahorse results for non-mitochondrial OCR for cells treated with diluent or caffeine. (H-I) Supernatant accumulation of interleukin- (IL-) 6 (H) and CXCL1 (I). Results are representative of two independent experiments and displayed as mean + SEM with dots indicating replicate wells. * = p<0.05; ** = p<0.01; *** = p <0.001; **** = p < 0.0001 as determined by ANOVA with Sidak adjustment.

fibroblasts demonstrated: more glycolytic activity as measured by extracellular acidification rate (ECAR) in the Seahorse assay (Fig 4A); a significant reduction in mitochondrial ATP production (Fig 4B); a significant reduction in basal oxidative phosphorylation as measured by the oxygen consumption rate (OCR; Fig 4C); and a higher spare respiratory capacity (SRC; Fig 4D). Rather than impact glycolysis, caffeine further inhibited OCR and mitochondrial ATP production in keloid FB (Fig 4A–4E).

## Scratch closure in keloid fibroblasts was dependent on glycolysis

A targeted inhibitor of glycolysis, 2DG, inhibited scratch results in keloid, but not HV cell line fibroblasts (Fig 4F and 4G). While the chemical inhibitor of mitochondrial oxidative phosphorylation (OxPhos), rotenone, did not impact the scratch assay results in either cell type (Fig 4F). This suggests that while mitochondrial ATP production was diminished in keloid line FB, the impact on scratch closure was independent of its further inhibition. However, non-mitochondrial OCR was preserved in keloid FB and inhibited by caffeine (Fig 4G). 2DG and rotenone inhibited the production of IL-6 in both keloid and healthy fibroblasts (Fig 4H). However, only 2DG inhibited CXCL1 (Fig 4I) indicating a potential role of metabolism in IL-6 mediated inflammation and neutrophil recruitment beyond the reported connection between IL-6 and glycolysis [224].

## Allicin altered metabolism and scratch closure

In our literature review, Allicin was also identified as an inhibitor of wound healing [85] and subsequently revealed modulation of metabolism via JAK2/STAT3 [225]. Allicin did inhibit scratch closure in both HV and keloid fibroblasts in a dose dependent fashion (Fig 5A and 5B). At higher doses, allicin caused a greater degree of cellular detachment from the plate (Figs 5A and S2A). Allicin did not significantly impact ECAR (Fig 6C), SRC (Fig 6D), or mitochondrial ATP production (Fig 5E). However, allicin selectively inhibited basal OCR (Fig 6F) and non-mitochondrial OCR (Fig 6G) while shifting the ECAR-OCR ratio (Fig 5H) in keloid fibroblasts. Vimentin staining was similarly reduced in dose-response fashion (Fig 5I–5J). At moderate doses, the staining pattern and cell morphology became disordered in both HV and keloid cells (Fig 5J). Although inhibition of IL-6 occurred to a greater degree in keloid cells than HV (Fig 5K), no rescue of CXCL1 production was seen (Fig 5L).

## Shikonin inhibited keloid fibroblasts without impacting metabolism

Given the potential that metabolic alterations could inhibit keloid fibroblasts as seen with caffeine and allicin, we revisited our original 1,331 articles to search for pharmacologically available products that inhibited the scratch assay with known impacts on metabolism. As previously described, Shikonin inhibited EMT in cancer cell lines [226], regulated Warburg physiology in keloid fibroblasts, and improved outcomes in murine burn models [227]. However, in our analysis shikonin did not alter basal ECAR (Fig 6A), basal OCR (Fig 6B), or SRC (Fig 6C). Shikonin inhibited mitochondrial ATP production but not non-mitochondrial OCR in keloid cells (Fig 6D and 6E). Shikonin also partially normalized the ECAR-to-OCR ratio (Fig 6F). Despite the paucity of metabolic impacts, shikonin inhibited the scratch assay results in both HV and keloid FB (Fig 6G) and suppressed vimentin staining in HV, but not keloid, cells (Fig 6H). Although shikonin inhibited the overproduction of IL-6 in keloid cells (Fig 6I), it did not normalize the production of CXCL1 (Fig 6J). Table 2 summarizes the impacts of each treatment on the measured outcomes.

## Discussion

The scratch assay is a widely used in vitro tool for assessing EndoMT, all three types of EMT, and overall wound healing [9]. The potential difference in results between commercial grade and extracted chondroitin are representative of a limitation of scratch assay research using molecularly complex stimuli. While such distinctions may not limit therapeutic benefit of the product tested, it is often unclear if the results are specific to the exact stimuli used or if the effects would be shared within the stimuli's general category. Therefore, given that many

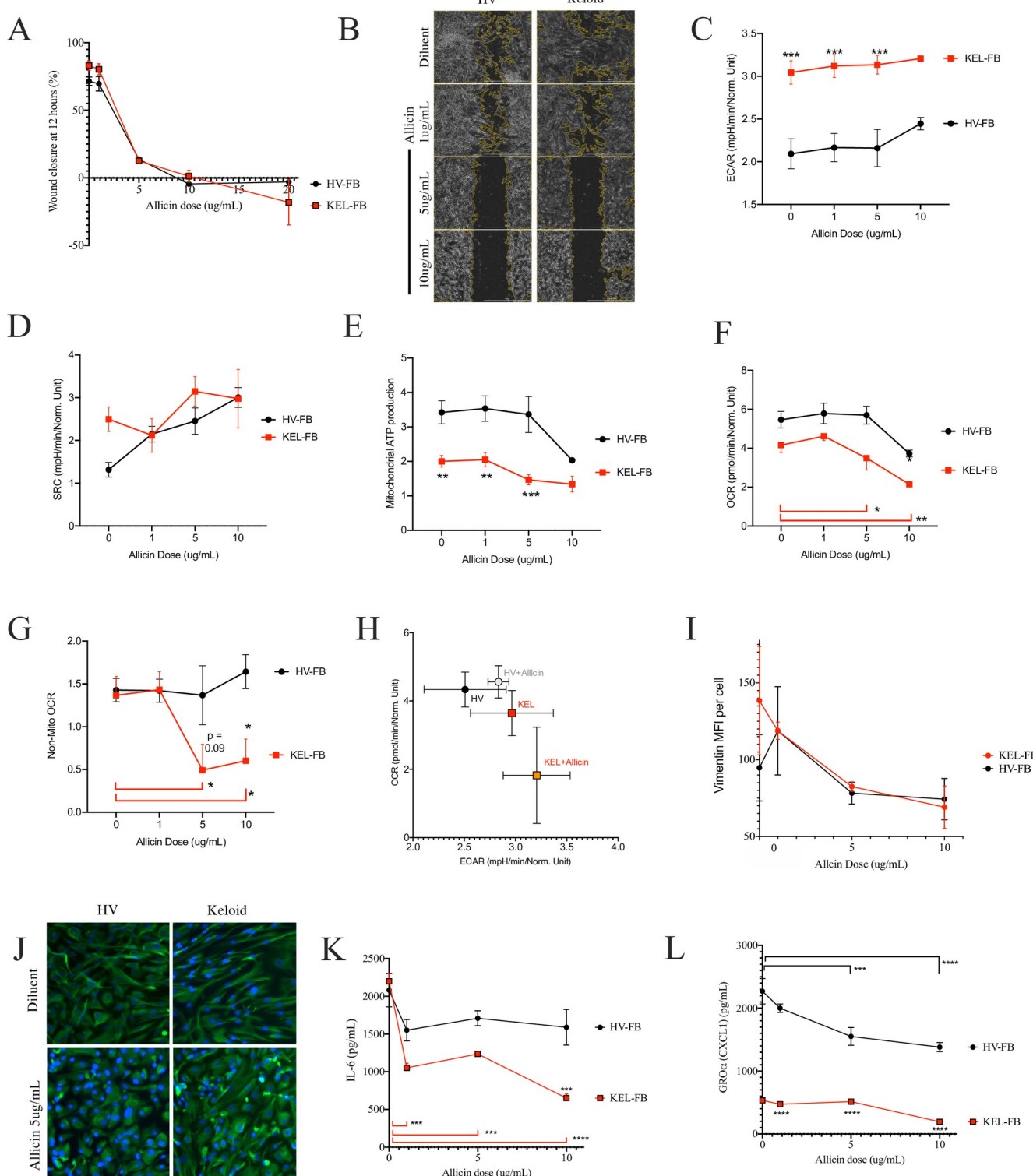

**Fig 5. Allicin impacts scratch outcomes and metabolic function.** (A-B) Wound closure as 12 hours (A) and representative image (B) for keloid (KEL-)or health volunteer (HV-) fibroblast cell lines (FB) after treatment with indicated doses of allicin. (C-F) Seahorse assay results for extracellular acidification rate (ECAR; C), spare respiratory capacity (SRC; D), mitochondrial (mito) ATP production (E), basal oxygen consumption rate (OCR; F), non-mitochondrial OCR (G), and ratio of basal ECAR to OCR (H) are shown. (I-H) Mean fluorescence intensity (MFI) per cell (I) and representative images (H) for vimentin for HV-FB and KEL-FB treated with indicated doses of allicin. (K-L) Supernatant accumulation of interleukin (IL-) 6 (K) and CXCL1 (L). Results are

representative of two independent experiments and displayed as mean + SEM (A-G, I-L) or SD (H) for triplicate wells. * = p<0.05; ** = p<0.01; *** = p <0.001; **** = p < 0.0001 as determined by ANOVA with Sidak adjustment compared with HV in under similar conditions unless indicated; red brackets indicate statistical assessment for keloid FB while black brackets represent statistical assessment for HC-FB. Masking on representative images of scratch assay performed by Scratch App (BioTek). Negative values on scratch healing indicate inhibition of scratch repair (healing times that were slower than diluent control).

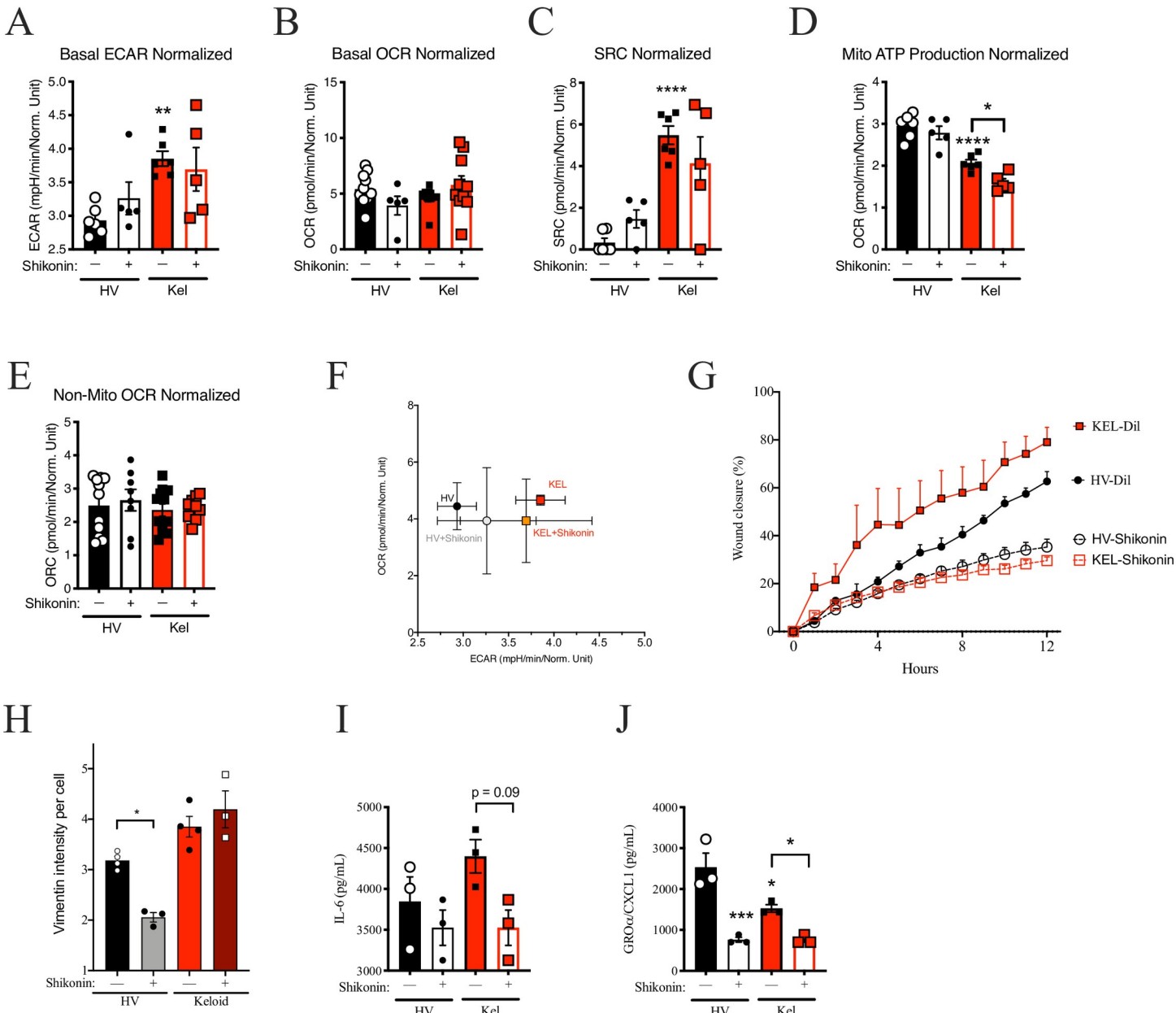

**Fig 6. Shikonin impacts scratch outcomes without influencing metabolic function.** (A-F) Seahorse assay results for keloid (KEL-) or health volunteer (HV-) fibroblasts (FB) for extracellular acidification rate (ECAR; A), basal oxygen consumption rate (OCR; B), spare respiratory capacity (SRC; C), mitochondrial (mito) ATP production (D), non-mitochondrial ATP production (E), and ratio of basal ECAR to OCR (F) are shown. (G) Wound closure as 12 hours for HV and KEL-FB after treatment with shikonin (10μM). (H) Mean fluorescence intensity (MFI) per cell for vimentin for HV-FB and KEL-FB treated with shikonin (10μM). (I-J) Supernatant accumulation of interleukin (IL-) 6 (I) and CXCL1 (J). Results are representative of three independent experiments and displayed as mean + SEM (A-E, G-J) or SD (F) for triplicate wells. * = p<0.05; ** = p<0.01; *** = p <0.001; **** = p < 0.0001 as determined by ANOVA with Sidak adjustment compared with HV in under similar conditions unless indicated.

**Table 2. Summary of impacts of compounds on healthy volunteer and keloid fibroblasts.** Summary of impacts of the glycolysis inhibitor (2DG), the mitochondrial ATP inhibitor (rotenone), caffeine, allicin, and shikonin on scratch assay healing time, extracellular acidification rate (ECAR), mitochondrial ATP production (Mito-ATP), non-Mito-ATP, the proinflammatory interleukin (IL-)6, and the neutrophil chemokine CXCL1.

| Stimulant | EMT | ECAR | Mito-ATP | Non-Mito-ATP | IL-6 | CXCL1 |
|---|---|---|---|---|---|---|
| **Healthy volunteer fibroblast cell line** | | | | | | |
| 2DG | — | ↓ | — | ↓ | ↓ | ↓ |
| Rotenone | — | — | ↓ | — | ↓ | — |
| Caffeine | ↓ | — | ↓ | ↓ | — | — |
| Allicin | ↓ | — | ↓ | ↓ | ↓ | ↓ |
| Shikonin | ↓ | — | — | — | — | ↓ |
| **Keloid fibroblast cell line** | | | | | | |
| Keloid derived (vs HV-FB) | ↑ | ↑ | ↓ | — | ↑ | ↓ |
| 2DG | ↓ | ↓ | — | ↓ | ↓ | ↓ |
| Rotenone | — | — | ↓ | — | ↓ | — |
| Caffeine | ↓ | — | ↓ | ↓ | — | — |
| Allicin | ↓ | — | ↓ | ↓ | ↓ | ↓ |
| Shikonin | ↓ | — | ↓ | — | ↓ | ↓ |

papers also failed to compare their findings against more than one cell type, researchers may have difficulty extrapolating findings beyond the exact parameters of the presented experiment. However, such research may occur prior to 2016 or have been performed using the transwell assay rather than scratch assay [221].

Our findings are also limited in the sole use of monolayer cultures of fibroblasts as the primary focus of the complex pathology of keloid scarsp. In addition, although the use of a commercial cell line allows other researchers greater opportunity for testing the reproducibility of our findings, our results are nonetheless limited to singular cell lines and thus cannot comment on the impact of body site or individual patient variation in healthy donors or patients with keloids. Furthermore, many scratch assay methods employ anti-proliferative agents to limit the interpretation of scratch closure results to migration. By avoiding use of these agents our results better reflect *in vivo* wound closure, which relies on both cell proliferation and migration; however, failure to use anti-proliferative treatments in our methods precludes us from commenting on whether the results seen on scratch closure were due to impacts on proliferation, migration, or both.

Despite limitations of the literature and our assay, we identified three over-the-counter treatments that improved modeled outcomes in the commercially available keloid fibroblast cell line: caffeine, allicin, and subsequently shikonin. Caffeine is available over the counter in creams marketed for reducing "cellulite" and diminishing "bags under the eyes". However, our results suggest that caffeine may be less ideal due to an increased potency inhibiting wound healing in healthy cell line FB than those from keloids (Fig 2E) and failed to impact the hyperinflammatory state of keloid line FB (Fig 3F). Allicin, a sulfur containing metabolite extracted from garlic, has also been used to mitigate murine models of fibrotic disorders like keloid scars and pulmonary fibrosis [85,225,228]. Allicin may present the most promising candidate for clinical trials given it appeared more potent against keloid FB than healthy cells in its OxPhos inhibition (Fig 5F), cell toxicity (Figs 5A and S2A), and IL-6 inhibition (Fig 5K). Shikonin has long been a traditional Chinese medicine with anti-scaring claims [227]. While shikonin inhibited scratch closure (Fig 6G) and IL-6 production (Fig 6I) in keloid line cells, the mechanism of action is unclear given shikonin failed to impact vimentin expression (Fig 6H) and had less pronounced impacts on metabolism (Fig 6A–6E).

However, the impacts of allicin and caffeine on OxPhos worsened, rather than reverse, the inherent mitochondrial ATP defect in keloid cells (Figs 4B, 4C, 5E and 5F). The reduction in non-mitochondrial OCR seen with caffeine and allicin treatment may indicate a role for reactive oxygen species and/or NOX mediated metabolism in the pathogenesis of keloids [229]. Thus, elucidation of the maladaptive impact of Warburg metabolism in keloid cells is essential to discover a treatment that reverses the underlying metabolic disorder in keloid derived cells. Furthermore, given that all of our identified treatments also inhibited wound closure in healthy FB and could theoretically prevent normal wound healing, each should likely be avoided in the immediate aftermath of an injury or surgery.

While our evaluations were successful in identifying commonly available drugs with therapeutic potential in cell models of keloid scars, we could not uncover a unifying mechanism for their actions. Caffeine and allicin may work through reducing mitochondrial ATP production and non-mitochondrial OCR beyond what the cell can tolerate. Meanwhile, 2DG blocks the glycolytic pathway that keloid cells are programed to prefer. However, shikonin also inhibited scratch closure without similar impacts on metabolism. Adding further complexity, reductions in IL-6 did not correlate with any of the identified metabolic alterations. Furthermore, while most keloid histology does not indicate a strong role for neutrophils, the stark reduction in neutrophil chemokines like CXCL1 may indicate a role for neutrophils early in the tissue repair pathology such as seen in other disorders [230,231]. Overall, our results suggest that the scratch assay is a valuable research tool but the current literature limits extrapolation between research groups' findings. Despite these limitations, our results support the consideration of clinical trials investigating use of available wound healing inhibitors, most reasonably allicin, in the treatment and/or prevention of keloid scars.

## Supporting information

**S1 Fig.** (A-D) Supernatant accumulation of HGF (A), CCL2 (B), VEGF (C) and PDGF (D) for HV-FB and KEL-FB stimulated with indicated doses of caffeine. Results are representative of three independent experiments and displayed as mean + SEM for triplicate wells. **** = $p < 0.0001$; for statistical comparison of area under the curve versus HV-FB under same stimulation conditions as determined by ANOVA with Sidak adjustment.
(PDF)

**S2 Fig.** (A) Representative image for HV and KEL-FB treated with 20mg/mL allicin for 12 hours. (B) Representative images for HV and KEL-FB cells treated with indicated doses of allicin and stained for vimentin (green) and DAPI (blue). Results are representative of two independent experiments.
(PDF)

## Author Contributions

**Conceptualization:** Mohammadali E. Alishahedani, Katelyn J. McCann, Ian A. Myles.

**Data curation:** Mohammadali E. Alishahedani, Manoj Yadav, Katelyn J. McCann, Ian A. Myles.

**Formal analysis:** Mohammadali E. Alishahedani, Manoj Yadav, Katelyn J. McCann, Ian A. Myles.

**Funding acquisition:** Ian A. Myles.

**Investigation:** Mohammadali E. Alishahedani, Manoj Yadav, Katelyn J. McCann, Portia Gough, Carlos R. Castillo, Jobel Matriz, Ian A. Myles.

**Methodology:** Mohammadali E. Alishahedani, Manoj Yadav, Katelyn J. McCann, Portia Gough, Carlos R. Castillo, Jobel Matriz, Ian A. Myles.

**Project administration:** Ian A. Myles.

**Resources:** Ian A. Myles.

**Supervision:** Manoj Yadav, Portia Gough, Ian A. Myles.

**Validation:** Manoj Yadav, Portia Gough, Ian A. Myles.

**Visualization:** Mohammadali E. Alishahedani, Katelyn J. McCann, Ian A. Myles.

**Writing – original draft:** Mohammadali E. Alishahedani, Manoj Yadav, Ian A. Myles.

**Writing – review & editing:** Mohammadali E. Alishahedani, Manoj Yadav, Katelyn J. McCann, Portia Gough, Carlos R. Castillo, Ian A. Myles.

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
