## [Decision Letter · Decision Letter 0]

12 May 2021

PONE-D-21-08934

Therapeutic candidates for keloid scars identified by qualitative review of scratch assay research for wound healing

PLOS ONE

Dear Dr. Myles,

Thank you for submitting your manuscript to PLOS ONE. After careful consideration, we have decided that your manuscript does not meet our criteria for publication and must therefore be rejected.

I am sorry that we cannot be more positive on this occasion, but hope that you appreciate the reasons for this decision.

Yours sincerely,

David G. Greenhalgh, MD

Academic Editor

PLOS ONE

Additional Editor Comments (if provided):

The reviewers have not provided a high enough score for acceptance.

Reviewers' comments:

Reviewer's Responses to Questions

**Comments to the Author**

1. Is the manuscript technically sound, and do the data support the conclusions?

Reviewer #1: No

Reviewer #2: Yes

2. Has the statistical analysis been performed appropriately and rigorously? 

Reviewer #1: No

Reviewer #2: Yes

3. Have the authors made all data underlying the findings in their manuscript fully available?

Reviewer #1: Yes

Reviewer #2: Yes

4. Is the manuscript presented in an intelligible fashion and written in standard English?

Reviewer #1: No

Reviewer #2: Yes

5. Review Comments to the Author

Reviewer #1: The authors hypothesize that agents that can inhibit migration in an in vitro scratch assay may represent potential anti-keloid therapeutics. Because keloid fibroblasts have been reported to exhibit increased rates of migration compared with normal skin fibroblasts, this is not an unreasonable assertion. However, it is unclear why the authors chose to emphasize the process of EMT, which involves much more than migration. Although partial EMT has been observed in keloid keratinocytes, there has been no definitive demonstration of EMT in keloids to date; that is, there is no evidence that keloid keratinocytes transition to mesenchymal cells. The authors describe EMT, and state that “Although EMT is definitionally limited to epithelial derived cells such as keratinocytes (KC),”. Yet, later in the manuscript they state that “given that all of our identified treatments also inhibited EMT in healthy FB and could theoretically prevent normal wound healing…”. This suggests an incomplete understanding of EMT (which is not clearly explained in the manuscript). Fibroblasts are mesenchymal and thus do not undergo EMT. Further, EMT is much more than simply increased migration.

The authors incorrectly state that the scratch assay is used to “analyze cell migration, proliferation, and cell-to-cell interaction.” The in vitro scratch assay is used to analyze migration rates in vitro. When properly done, the cells are growth-inactivated to eliminate proliferation as a confounding variable; a scratch assay is not used to measure proliferation. It is not commonly used to study cell-to-cell interactions as it involves a single cell type.

The authors are relatively critical of many of the previous reports they cite, finding deficiencies in previous reports and contrasting those with the current study. However, their study suffers from many of the same limitations. For example, they failed to growth-inactivate the fibroblasts prior to scratching, which introduces proliferation as a confounding variable. They claim that this was done to “better reflect in vivo wound closure,” but it is not clear that this is the case. The scratch assay is one tool that investigators can use to study cell behavior, but the results of these assays must be interpreted in the limited context in which the assays are performed. Among the weaknesses in previously published scratch assays described by the authors, a major weakness was the use of diluent as a control. It is not clear how “competing serum” would serve as a control. I disagree that a “competing challenge of similar, but distinct, molecular complexity” is required for the assay to be valid.

The methods for cell culture must be described in detail. Did the medium contain serum? A reference for cell culture was provided, but some details must be provided here, including the culture medium. If it contained serum, as suggested by reference #8, this could have a tremendous impact on the results of the assay.

For the scratch assay, the methods describe “100,000-150,000 cells were added” per well in a 24-well plate; thus, one well may have had 50% cells more than another well. Differences in cell density can have a large effect in the scratch assay. Similarly, the methods state that cells were scratched “12-24 hours later,” but this variability can have significant effects since fibroblasts divide every 16-24 hours. This is a particular problem because the cells were not growth-inactivated. The authors must address these potentially confounding factors.

A major concern is that the authors used only a single donor strain for each fibroblast type, keloid or normal, making it impossible to draw broad conclusions regarding keloid pathology. There can be significant donor-to-donor variability in primary fibroblasts. Without studying multiple donors of each cell type, we cannot know whether any observed differences are due to this variability or are directly related to keloid pathology. This limitation must be addressed.

The Introduction should be used to provide background information and give context to the experiments reported in the manuscript. Instead, it appears the authors summarized the results of the current study in the second paragraph of the Introduction. This is not appropriate, and may be confusing to readers. They begin this paragraph by discussing “the scratch assay literature,” but the lack of citations suggests that the authors are describing their own results. A brief summary of the results at the end of the Introduction is not unusual, but this section is not the place to focus on the results of the current study.

The authors state, in the Methods: “Cell lines were not collected for this study, were collected through medically prescribed processes, and were completely de-identified to the researchers before access”. It is not clear why this was included as it appears the authors only used cells that were obtained from commercial sources. Note that primary cells are distinct from “cell lines.”

All abbreviations must be defined on first use (e.g., PFA, PBS, aSMA).

In the Methods for the Seahorse assay, the authors state “10,000 fibroblasts from healthy volunteers, keloid patients were seeded in 96-well cell culture ….” This sentence is confusing, as it appears that only a single donor cell strain for each, normal or keloid fibroblast, were used.

Alpha-SMA is not a “known EMT modulator” (line 185). It is often referred to as a marker of EMT, but not an EMT modulator. The cited paper (#14) does not discuss EMT, but uses a-SMA as a marker of vascular smooth muscle cells.

For Figure 2, why do the plots in A and D go out to only 16 hours, while the plots in E and G do extend to 20 hours (and, data for 20 hours for CS are plotted as a bar plot separate from the data shown in A)? The data should all be presented clearly and consistently. Further, why did they only analyze chondroitin in HV fibroblasts but caffeine was analyzed in both HV and keloid fibroblasts?

What was the purpose of analyzing vimentin staining? More justification should be given. The authors state that “Consistent with prior reports in KC [19], keloid FB had significantly more immunofluorescent staining per cell for the mesenchymal phenotype marker vimentin.” They provided a single reference, which is inconsistent with “previous reports” (plural), and the citation describes a study that involved keratinocytes, not fibroblasts. There are many more relevant markers of keloid fibroblasts that could have been used to demonstrate an effect on the keloid phenotype. In addition, more information about how vimentin staining was quantified should be provided. How was this controlled? How many replicates were performed?

For the protein expression data shown in Figure 3, the protein concentrations are expressed as pg/ml rather than referencing protein secretion to cell number. Without knowing the effects of caffeine on proliferation, and without demonstrating a control for cell number, the validity of the data is called into question. The differences in protein concentration per ml may be influenced by cell density.

Reviewer #2: The authors performed a qualitative review of the recent literature searching for inhibitors of scratch assay activity that where already available in topical formulations under the hypothesis that such compounds may offer therapeutic potential in keloid treatment. Caffeine and allicin inhibited the scratch assay closure and inflammatory abnormalities in the commercially available keloid fibroblast cell line. Caffeine and allicin also impacted ATP production in keloid cells, most notably an inhibition of non-mitochondrial oxygen consumption. The traditional Chinese medicine, shikonin, was also successful in inhibiting scratch closure but displayed less dramatic impacts on metabolism. Their results summarize the strengths and limitations of current scratch assay literature and suggest clinical assessment of the therapeutic potential for these identifiedcompounds against keloid scars may be warranted.

Overall,the manuscript is well written and has a few minor diction and typographic errors. The topic is important to wound healing clinicians and investigators.

Concerns with the manuscript:

1. A large amount of information is conveyed in the manuscript which may be excessive in terms of the number of references, the usefulness of Table 1. Effect of Different Stimuli on Scratch Assay in Keratinocytes, Fibroblasts, and Epithelial Cells which summarizes many drugs, agents and other cell types tested in the scratch assay. It may be more appropriate for this information to be conveyed in the supplementary data section.

2. In figure 2 C and F, it is unclear what timepoint was measured.

3. In figure 2 H, Figure 3 H and I, the error bars are not apparent and ** appear to represent stat sig but what values and what concentration of caffeine?

4. In figure 5 negative values for % of wound closure appear in part A which is difficult to interpret.

5. Conceptually, the entire manuscript deals with in vitro data and the scratch assay. The limitation of in vitro data to the in vivo reality are very large and complex although acknowledged no solution to overcome the applicability of the data to the in vivo circumstance is apparent.

6. PLOS authors have the option to publish the peer review history of their article (what does this mean?). If published, this will include your full peer review and any attached files.

Reviewer #1: No

Reviewer #2: No

- - - - -

---

## [Author Response · Author response to Decision Letter 0]

22 May 2021

Reviewer #1: The authors hypothesize that agents that can inhibit migration in an in vitro scratch assay may represent potential anti-keloid therapeutics. Because keloid fibroblasts have been reported to exhibit increased rates of migration compared with normal skin fibroblasts, this is not an unreasonable assertion. However, it is unclear why the authors chose to emphasize the process of EMT, which involves much more than migration. Although partial EMT has been observed in keloid keratinocytes, there has been no definitive demonstration of EMT in keloids to date; that is, there is no evidence that keloid keratinocytes transition to mesenchymal cells. The authors describe EMT, and state that “Although EMT is definitionally limited to epithelial derived cells such as keratinocytes (KC),”. Yet, later in the manuscript they state that “given that all of our identified treatments also inhibited EMT in healthy FB and could theoretically prevent normal wound healing…”. This suggests an incomplete understanding of EMT (which is not clearly explained in the manuscript). Fibroblasts are mesenchymal and thus do not undergo EMT. Further, EMT is much more than simply increased migration.

 We previously submitted this manuscript to a different journal and got this reviewer. As was detailed to the editor, it is clear this reviewer did not bother to read the new version (which incorporated their comments from the prior review). Instead this reviewer only cut and pasted their past review here, including quotes from the prior version. The lack of professionalism should invalidate their opinion, but if they are firm that copying past reviews without reading the updated manuscript is defensible they should identify themselves rather than half-heartedly performing reviews under the cover of blinding. We are very clear in this version that we are speaking of only proliferation and migration. This one use can be changed to “proliferation and migration” to match but the intro and discussion are clear we are referencing proliferation and migration. The PRIOR version, we use EMT in places we now refer to proliferation and migration. We will complete the point by point as requested but if Plos One has been provided evidence of a review performed in bad faith (misquoting the paper and misrepresenting it as if it were still not amended) then we have not further interest. 

The authors incorrectly state that the scratch assay is used to “analyze cell migration, proliferation, and cell-to-cell interaction.” The in vitro scratch assay is used to analyze migration rates in vitro. When properly done, the cells are growth-inactivated to eliminate proliferation as a confounding variable; a scratch assay is not used to measure proliferation. It is not commonly used to study cell-to-cell interactions as it involves a single cell type.

The reviewer appears to be making the assertion that cell-to-cell interactions cannot be performed in a culture involving a single cell type? We detail specifically that the scratch is typically used for migration, and that our results cannot separate out migration from proliferation. Admittedly, there may be an International Society for Scratch Assay Scientists consensus statement mandating anti-proliferative treatment we are unaware of. Short of that, the reviewer is confusing “tradition” with validity. The argument advanced of “this is how it is usually done” does not mean much if one is trying to find something new. We ran an assay for which proliferation and migration may contribute to results, we explain clearly that both are involved and why we did not want to limit to only one. But to do otherwise risks missing any potential treatment for keloids that only works on proliferation. We realize the reviewer has uncovered countless novel treatment options during using a tireless adherence to prior dogma, yet we naively hope that trying something new might lead to our own novel outcomes. 

The authors are relatively critical of many of the previous reports they cite, finding deficiencies in previous reports and contrasting those with the current study. However, their study suffers from many of the same limitations. For example, they failed to growth-inactivate the fibroblasts prior to scratching, which introduces proliferation as a confounding variable. They claim that this was done to “better reflect in vivo wound closure,” but it is not clear that this is the case. 

No one could rationally imply normal wound healing in vivo only involves cell migration. This would mean a scraped knee would heal by recruiting cells from elsewhere and a “road rash” wound from a motorcycle accident that scraped up the patients entire side would require other areas of the body to denude in order to heal. Clearly if one wanted to make a pure migration assay, one could use the scratch assay and add an anti-proliferative agent. But to test full capacity for wound closure, both proliferation and migration are more in line with in vivo pathways. Again we are clear that we can’t know of the identified treatments block migration, proliferation, or both. But the reviewer provides no reasoning as to why the therapeutic potential is invalidated by proliferation being a factor. 

The scratch assay is one tool that investigators can use to study cell behavior, but the results of these assays must be interpreted in the limited context in which the assays are performed. Among the weaknesses in previously published scratch assays described by the authors, a major weakness was the use of diluent as a control. It is not clear how “competing serum” would serve as a control. I disagree that a “competing challenge of similar, but distinct, molecular complexity” is required for the assay to be valid.

We clearly state that the paper in question used alligator serum versus water and then made very specific claims about the ability of alligator serum to induce EMT. However, we are noting that what is unclear is whether or not the properties are unique to alligator serum, or just any serum. Or just nutrients for that matter. A finding of “a nutrient rich media helped cells grow/move better than a nutrient poor media” is all adding serum can say. Having one condition where cells are grown in a low nutrient media, and one where than have nigh nutrients is not a comment on the source of those nutrients. We admit “serum starvation” is a common practice in cell culture work, the results of which imply unique properties to serum. However, the authors of the crocodile serum paper did not test if the effect could be explained by simply increasing the nutrients in the competing serum (ie comparing a media with equivalent macromolecule energy content that were derived from non-serum sources). If I compared one person under fasting conditions and another eating Oreos, would I be able to claim that Oreos uniquely caused insulin to increase? We have added to the paper a discussion that practically it may not matter if the goal is to induce wound healing. However mechanistic conclusions are easy for papers where the only variable is the addition of one cytokine or molecule versus studies that use a highly complex stimulant versus water or saline. 

The methods for cell culture must be described in detail. Did the medium contain serum? A reference for cell culture was provided, but some details must be provided here, including the culture medium. If it contained serum, as suggested by reference #8, this could have a tremendous impact on the results of the assay.

The cited paper clearly outlines how we cultured our cells. 

For the scratch assay, the methods describe “100,000-150,000 cells were added” per well in a 24-well plate; thus, one well may have had 50% cells more than another well. Differences in cell density can have a large effect in the scratch assay. Similarly, the methods state that cells were scratched “12-24 hours later,” but this variability can have significant effects since fibroblasts divide every 16-24 hours. This is a particular problem because the cells were not growth-inactivated. The authors must address these potentially confounding factors.

Had the review read (and quoted) the full sentence of this paper rather the prior version, their inquiry would have been answered. The reviewer states “For the scratch assay, the methods describe “100,000-150,000 cells were added” per well in a 24-well plate; thus, one well may have had 50% cells more than another well.” We clarified in the PLOS One version (again, in the second half of the sentence) that we “(cell number was matched across conditions within each given experiment)”. That is clearly stated but neglected by this reviewer. So no, it was not that one well might have 50% more than the other it was that between different experiments we might have differences in seeding densities. The fact that we still found impacts within each experiment despite variance across experiments enhances our argument. For the reviewer to misrepresent our method to this degree means they either did not make a good faith effort to read the new manuscript or they are intentionally acting in bad faith. 

A major concern is that the authors used only a single donor strain for each fibroblast type, keloid or normal, making it impossible to draw broad conclusions regarding keloid pathology. There can be significant donor-to-donor variability in primary fibroblasts. Without studying multiple donors of each cell type, we cannot know whether any observed differences are due to this variability or are directly related to keloid pathology. This limitation must be addressed.

We are very clear in the discussion that the study is limited to one cell line and the reasons why we still find this valuable. We outline that the next step would include testing a larger number of cells. Obviously human samples will be variable but a researcher with access to large cell banks would likely read our work and decide to try the same process for themselves. To inspire attempts at reproducibility is not a knock on a paper so long as it clearly articulates it’s own shortfalls. Initiating a clinical trial to collect numerous samples to further test our findings would be a massive investment of time and money; if the reviewer is imply our findings warrant this type of dedicated program for sample collection and testing, they are making a rather strong endorsement of our paper rather than refuting it. 

The Introduction should be used to provide background information and give context to the experiments reported in the manuscript. Instead, it appears the authors summarized the results of the current study in the second paragraph of the Introduction. This is not appropriate, and may be confusing to readers. They begin this paragraph by discussing “the scratch assay literature,” but the lack of citations suggests that the authors are describing their own results. A brief summary of the results at the end of the Introduction is not unusual, but this section is not the place to focus on the results of the current study.

This is not the type of thing objective reviewers obsess over, but it is good to know the reviewer at least read the intro. 

The authors state, in the Methods: “Cell lines were not collected for this study, were collected through medically prescribed processes, and were completely de-identified to the researchers before access”. It is not clear why this was included as it appears the authors only used cells that were obtained from commercial sources. Note that primary cells are distinct from “cell lines.”

This statement was specifically required by PLOS One prior to consideration.

All abbreviations must be defined on first use (e.g., PFA, PBS, aSMA).

This has been amended. 

In the Methods for the Seahorse assay, the authors state “10,000 fibroblasts from healthy volunteers, keloid patients were seeded in 96-well cell culture ….” This sentence is confusing, as it appears that only a single donor cell strain for each, normal or keloid fibroblast, were used.

We clarified one cell line per group when updating this paper from the prior version. 

Alpha-SMA is not a “known EMT modulator” (line 185). It is often referred to as a marker of EMT, but not an EMT modulator. The cited paper (#14) does not discuss EMT, but uses a-SMA as a marker of vascular smooth muscle cells.

We do not use the phrase “known EMT modulator” in this version, only in the prior. This version says “a potential EMT modulator”. If PLOS One is okay with this behavior let’s arrange a Zoom to chat about it without allowing bad faith efforts to hide behind blinded processes. 

For Figure 2, why do the plots in A and D go out to only 16 hours, while the plots in E and G do extend to 20 hours (and, data for 20 hours for CS are plotted as a bar plot separate from the data shown in A)? The data should all be presented clearly and consistently. Further, why did they only analyze chondroitin in HV fibroblasts but caffeine was analyzed in both HV and keloid fibroblasts?

The paper specifically discusses that we did not carry forward with chondroitin because we could not find evidence of activity in HV cells.

What was the purpose of analyzing vimentin staining? More justification should be given. The authors state that “Consistent with prior reports in KC [19], keloid FB had significantly more immunofluorescent staining per cell for the mesenchymal phenotype marker vimentin.” They provided a single reference, which is inconsistent with “previous reports” (plural), and the citation describes a study that involved keratinocytes, not fibroblasts. There are many more relevant markers of keloid fibroblasts that could have been used to demonstrate an effect on the keloid phenotype. In addition, more information about how vimentin staining was quantified should be provided. How was this controlled? How many replicates were performed?

This is in the methods section.

For the protein expression data shown in Figure 3, the protein concentrations are expressed as pg/ml rather than referencing protein secretion to cell number. Without knowing the effects of caffeine on proliferation, and without demonstrating a control for cell number, the validity of the data is called into question. The differences in protein concentration per ml may be influenced by cell density.

As above, every well in a given experiment is already adjusted for cell density by the fact that the same numbers of cells are plates per condition. 

Reviewer #2: The authors performed a qualitative review of the recent literature searching for inhibitors of scratch assay activity that where already available in topical formulations under the hypothesis that such compounds may offer therapeutic potential in keloid treatment. Caffeine and allicin inhibited the scratch assay closure and inflammatory abnormalities in the commercially available keloid fibroblast cell line. Caffeine and allicin also impacted ATP production in keloid cells, most notably an inhibition of non-mitochondrial oxygen consumption. The traditional Chinese medicine, shikonin, was also successful in inhibiting scratch closure but displayed less dramatic impacts on metabolism. Their results summarize the strengths and limitations of current scratch assay literature and suggest clinical assessment of the therapeutic potential for these identified compounds against keloid scars may be warranted.

Overall, the manuscript is well written and has a few minor diction and typographic errors. The topic is important to wound healing clinicians and investigators.

Concerns with the manuscript:

1. A large amount of information is conveyed in the manuscript which may be excessive in terms of the number of references, the usefulness of Table 1. Effect of Different Stimuli on Scratch Assay in Keratinocytes, Fibroblasts, and Epithelial Cells which summarizes many drugs, agents and other cell types tested in the scratch assay. It may be more appropriate for this information to be conveyed in the supplementary data section.

It is a long table, but given the online format of PLOS One we did not think that would be a problem given the focus of the paper is very much in Table 1. But we would have no issues moving this to supplemental should the editors agree or reviewer feel strongly.

2. In figure 2 C and F, it is unclear what timepoint was measured.

C was clarified as 18 hours, F was already listed as 22 hours. Thank you for catching this oversite. 

3. In figure 2 H, Figure 3 H and I, the error bars are not apparent and ** appear to represent stat sig but what values and what concentration of caffeine?

We have attempted to better emphasize that the figure legend explains the stats indicate the area under the curve. No error bars would be present in 2H because it was a average at different doses, stats calculated on area under the curve. 3H and I the error bars a present but small. 

4. In figure 5 negative values for % of wound closure appear in part A which is difficult to interpret.

We have attempted to clarify that negative values mean that the compound added at that concentration was inhibitory. Comparisons are made to diluent control, thus an inhibitor of closure would have a negative value.

5. Conceptually, the entire manuscript deals with in vitro data and the scratch assay. The limitation of in vitro data to the in vivo reality are very large and complex although acknowledged no solution to overcome the applicability of the data to the in vivo circumstance is apparent.

We agree, but the only wat to move to humans in a clinical trail. We are actively planning such a trial, but in order to justify it think publishing the underlying data is appropriate.

---

## [Decision Letter · Decision Letter 1]

10 Jun 2021

Therapeutic candidates for keloid scars identified by qualitative review of scratch assay research for wound healing

PONE-D-21-08934R1

Dear Dr. Myles,

We’re pleased to inform you that your manuscript has been judged scientifically suitable for publication and will be formally accepted for publication once it meets all outstanding technical requirements.

Kind regards,

Michael Klymkowsky, Ph.D.

Academic Editor

PLOS ONE

Additional Editor Comments (optional):

Reviewers' comments:

Reviewer's Responses to Questions

**Comments to the Author**

1. If the authors have adequately addressed your comments raised in a previous round of review and you feel that this manuscript is now acceptable for publication, you may indicate that here to bypass the “Comments to the Author” section, enter your conflict of interest statement in the “Confidential to Editor” section, and submit your "Accept" recommendation.

Reviewer #2: All comments have been addressed

Reviewer #3: All comments have been addressed

2. Is the manuscript technically sound, and do the data support the conclusions?

Reviewer #2: Yes

Reviewer #3: Yes

3. Has the statistical analysis been performed appropriately and rigorously? 

Reviewer #2: Yes

Reviewer #3: I Don't Know

4. Have the authors made all data underlying the findings in their manuscript fully available?

Reviewer #2: Yes

Reviewer #3: Yes

5. Is the manuscript presented in an intelligible fashion and written in standard English?

Reviewer #2: Yes

Reviewer #3: Yes

6. Review Comments to the Author

Reviewer #2: There authors have addressed the concerns expressed. Given the on line format of the journal, the inclusion of the table one in the manuscript is acceptable as suggested by the authors.

However, the manuscript still has 231 references which for most publications is excessive. This editorial discretion is required to decide to include these large bodies of information.

Reviewer #3: (No Response)

7. PLOS authors have the option to publish the peer review history of their article (what does this mean?). If published, this will include your full peer review and any attached files.

Reviewer #2: No

Reviewer #3: No

---

## [Editor Report · Acceptance letter]

11 Jun 2021

PONE-D-21-08934R1 

Therapeutic candidates for keloid scars identified by qualitative review of scratch assay research for wound healing 

Dear Dr. Myles:

I'm pleased to inform you that your manuscript has been deemed suitable for publication in PLOS ONE. Congratulations! Your manuscript is now with our production department. 

Kind regards, 

on behalf of

Dr. Michael Klymkowsky 

Academic Editor

PLOS ONE